# Vanishing white matter disease expression of truncated EIF2B5 activates induced stress response

Matthew D Keefe[1†], Haille E Soderholm[1†], Hung-Yu Shih[1†], Tamara J Stevenson[1], Kathryn A Glaittli[1], D Miranda Bowles[1], Erika Scholl[1], Samuel Colby[2], Samer Merchant[2], Edward W Hsu[2], Joshua L Bonkowsky[1,3]*

[1]Department of Pediatrics, University of Utah School of Medicine, Salt Lake City, United States; [2]Department of Bioengineering, University of Utah, Salt Lake City, United States; [3]Brain and Spine Center, Primary Children's Hospital, Salt Lake City, United States

**Abstract** Vanishing white matter disease (VWM) is a severe leukodystrophy of the central nervous system caused by mutations in subunits of the eukaryotic initiation factor 2B complex (eIF2B). Current models only partially recapitulate key disease features, and pathophysiology is poorly understood. Through development and validation of zebrafish (*Danio rerio*) models of VWM, we demonstrate that zebrafish *eif2b* mutants phenocopy VWM, including impaired somatic growth, early lethality, effects on myelination, loss of oligodendrocyte precursor cells, increased apoptosis in the CNS, and impaired motor swimming behavior. Expression of human *EIF2B2* in the zebrafish *eif2b2* mutant rescues lethality and CNS apoptosis, demonstrating conservation of function between zebrafish and human. In the mutants, intron 12 retention leads to expression of a truncated *eif2b5* transcript. Expression of the truncated *eif2b5* in wild-type larva impairs motor behavior and activates the ISR, suggesting that a feed-forward mechanism in VWM is a significant component of disease pathophysiology.

*For correspondence:
joshua.bonkowsky@hsc.utah.edu

†These authors contributed equally to this work

## Introduction

Vanishing white matter (VWM) disease is a genetic leukodystrophy leading to severe neurological disease and early death (*Fogli et al., 2004*; *van der Knaap et al., 2006*; *Labauge et al., 2009*; *Carra-Dalliere et al., 2011*; *VWM Research Group et al., 2018*). VWM disease is caused by bi-allelic recessive variants in any of the five genes encoding subunits (1-5) of the eukaryotic translation initiation factor 2B (eIF2B) complex. Symptoms of VWM include ataxia, spasticity, seizures, cognitive impairment, and motor problems. While there are no sex differences seen in VWM patients, females can experience ovarian failure. In patients, autopsies have revealed affected oligodendrocytes and astrocytes, myelin loss, and cystic cavitations of white matter (*van der Knaap et al., 1997*; *van der Knaap et al., 1998*; *Pronk et al., 2006*). Genotype-phenotype correlations of mutation severity have been shown (*VWM Research Group et al., 2018*), and severe multi-organ involvement is characteristic of fetal and neonatal forms (*van der Knaap et al., 2003b*; *Song et al., 2017*). Disease onset <4 years of age is followed by a rapid deterioration of symptoms, while disease onset >4 years of age shows greater variability in disease course (*VWM Research Group et al., 2018*). There is no treatment for VWM disease, and current mouse models only partially recapitulate disease pathophysiology (*Geva et al., 2010*; *Dooves et al., 2016*).

The eIF2B complex is a heteropentameric guanine nucleotide exchange factor (GEF) for eukaryotic initiation factor 2 (eIF2), which governs the rate of global protein synthesis and cap-dependent translation initiation. The eIF2B complex also functions to displace eIF5 from inactive GDP-bound

eIF2 to allow its recruitment to the ribosome (*Jennings and Pavitt, 2014*). Importantly, the eIF2B complex plays a central role in the cellular integrated stress response (ISR). Stress-dependent kinase activation leads to phosphorylation of eIF2, which binds eIF2B more tightly and reduces overall protein synthesis (*Krishnamoorthy et al., 2001*; *Pakos-Zebrucka et al., 2016*). In human cell lines, it has been shown that translation is suppressed to a greater degree after stress in VWM patients (*Moon and Parker, 2018*). This suppression of translation lasts for a longer period of time in VWM cells, and the protein responsible for de-phosphorylating eIF2 and allowing translation recovery, GADD34, was found in lower quantities.

Mouse lines with knock-in/knock-out mutations in *Eif2b5*[R132H], a common allele of VWM patients, have impaired motor function, growth deficits, delayed development of white matter, and abnormal abundance of oligodendrocytes and astrocytes (*Geva et al., 2010*; *Atzmon et al., 2018*). This developmental role for eIF2B was further confirmed in a mouse model with developmental misexpression of pancreatic endoplasmic reticulum kinase (PERK) in oligodendrocytes. PERK is one of the stress-responsive kinases that activate the ISR via phosphorylation of eIF2. PERK misexpression caused hypomyelination, oligodendrocyte damage, and myelin loss (*Lin et al., 2014*). However, this result was only seen in young mice, and could not be induced in mature animals.

ISR activation has been identified in VWM patient brain autopsy samples (*van Kollenburg et al., 2006*) and in mouse VWM models (*Wong et al., 2019*; *Abbink et al., 2019*). A small molecule inhibitor of the ISR, ISRIB, has been shown to bind to and activate the decameric, functional eIF2B complex (*Sidrauski et al., 2013*; *Sidrauski et al., 2015*; *Wong et al., 2018*).

There are key aspects that remain unclear about eIF2B function and its involvement in VWM pathophysiology. eIF2B is expressed globally but VWM primarily affect the CNS, including differential effects in the CNS. For example, oligodendrocyte numbers are decreased in affected white matter, but they are increased in other areas (*Van Haren et al., 2004*; *Bugiani et al., 2010*). GEF activity of the eIF2B complex does not appear to correlate with VWM disease severity (*Liu et al., 2011*), suggesting that the overall protein translation is not the key component of VWM pathophysiology. Another unusual and poorly understood aspect of VWM is that rapid clinical deterioration and white matter loss can be provoked by a stressor, such as minor head trauma or mild illness (*van der Knaap et al., 2006*). This is consistent with models in which chronic ISR activation causes cellular apoptosis (*Bond et al., 2020*), but the mechanism by which VWM mutation affects ISR response is unclear. Further, there is some evidence that a de-regulated ISR is not alone sufficient to cause VWM pathology (*Wisse et al., 2017*), and in fact blocking the ISR worsens VWM pathology (*Sekine et al., 2016*). Thus, current models and approaches have left key questions unanswered.

We report the development and characterization of a small vertebrate (zebrafish – *Danio rerio*) model of VWM. We have generated and characterized mutant alleles in zebrafish *eif2b* subunits 1, 2, and 4, and an allelic series in subunit 5. We demonstrate that the *eif2b* mutants exhibit a range of phenotypic severity, including changes in growth, lethality, behavior, myelination, apoptosis, and proliferation in the CNS. We also show conservation of function of *eif2b2* between zebrafish and humans, validating the zebrafish model for understanding human VWM. We find that the *eif2b* mutants at baseline have activated induced stress response (ISR). The zebrafish *eif2b* mutants have impaired swimming motor behavior, a phenotype that could be used in a phenotype-based screen. Finally, we found that in the zebrafish *eif2b* mutants, a truncated *eif2b5* transcript is generated. The truncated *eif2b5* is capable of activating the ISR, and can impair motor behavior, suggesting that a feedback loop with truncated *eif2b5* may play an important role in disease pathology. Our work reveals a similar role for the *eif2b* complex in zebrafish and in humans, identifies a novel mechanism of VWM pathophysiology, and provides a useful model of VWM for future therapeutics screening.

## Results

### Phylogenetic analysis of EIF2B sequence homology and expression in zebrafish development

We characterized the sequence and developmental expression of the five Eif2b subunit orthologs in zebrafish, *eif2b1-5* (*eif2b1*, ENSDARG00000091402; *eif2b2* ENSDARG00000041397; *eif2b3*, ENSDARG00000018106; *eif2b4*, ENSDARG00000014004; *eif2b5*, ENSDARG00000074995). Each human *EIF2B* subunit gene has a single zebrafish ortholog with a conserved amino acid sequence

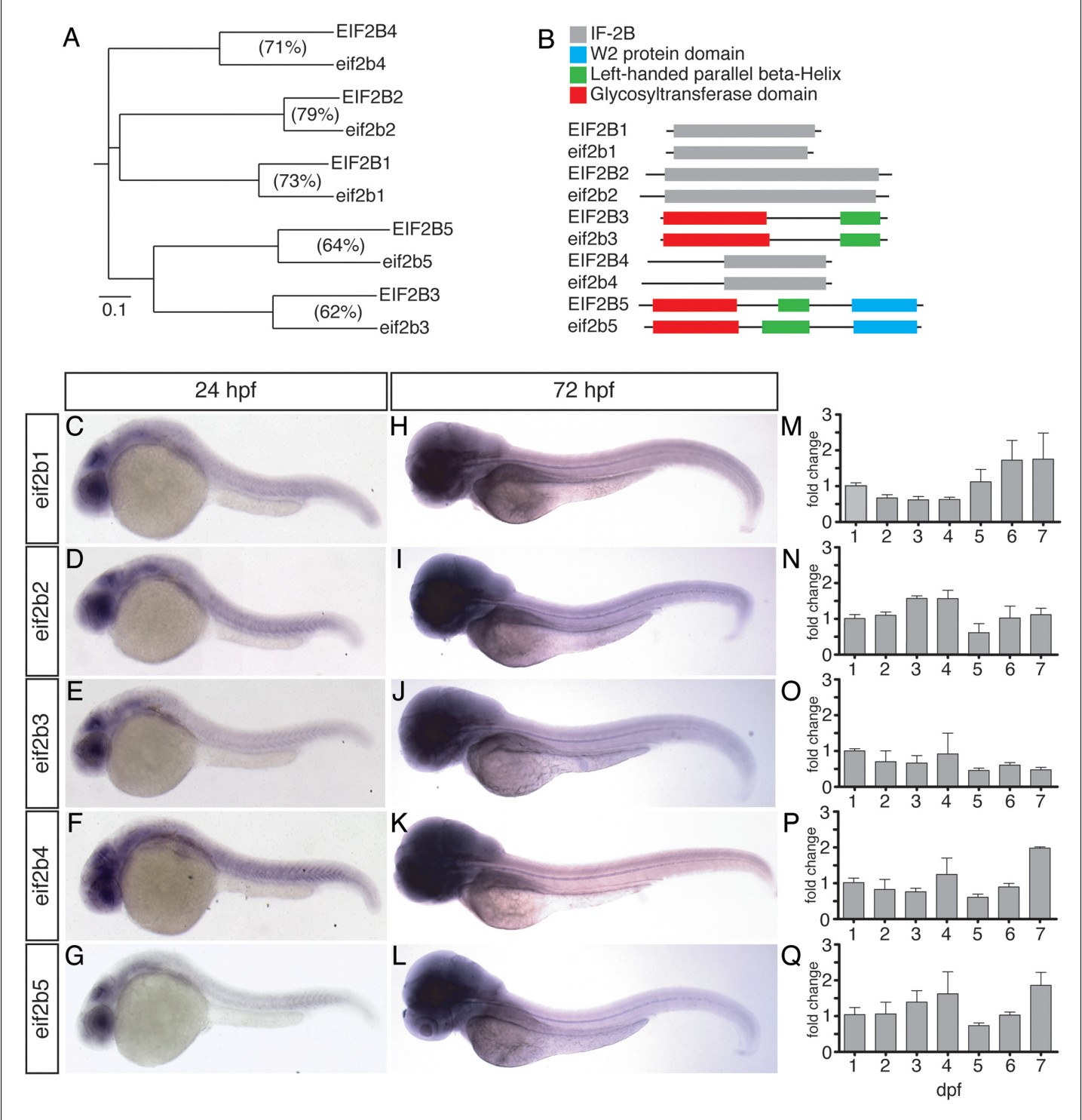

**Figure 1.** Phylogenetic and expression analysis of Eif2b orthologs in zebrafish during development. (A) A horizontal cladogram of *eif2b1-5* gene sequences shows that zebrafish have a single ortholog of each human *EIF2B* gene, and a relative conservation of amino acid sequence between orthologs. Scale bar equals an evolutionary distance of 0.1 amino acid changes per position in the sequence (Phylodendron). (B) Comparison of zebrafish and human EIF2B protein sequences shows conserved domain architectures. (C–L) Whole-mount in situ expression analysis in zebrafish embryos and larvae, brightfield microscopy, rostral left, dorsal top. (C–E) *eif2b* subunit genes at 24 hpf shows predominantly brain and eye expression, with lower levels of expression throughout the embryo. (H–L) *eif2b* subunit genes at 72 hpf shows higher expression throughout the animal and in the brain compared to 24 hpf. (M–Q) qRT-PCR expression of *eif2b* subunit genes from 24 hpf through 7 dpf, normalized to 24 hpf expression demonstrates variable expression changes across development. Error bars, standard error of the mean.

*Figure 1 continued on next page*

*Figure 1 continued*

The online version of this article includes the following source data for figure 1:

**Source data 1.** Quantification of qRT-PCR results of *eif2b* subunits.
**Source data 2.** Statistical analysis of changes in subunit expression.

(*Figure 1A*). The Eif2b zebrafish and human protein structures also show conserved structural domains (*Figure 1B*). We evaluated *eif2b* gene expression during early development. Expression analysis of zebrafish *eif2b* genes at 24 hpf (hours post-fertilization) showed expression chiefly in the brain and eye (*Figure 1C–G*). By 72 hpf expression was clearly noted throughout the animal, with higher levels in the brain (*Figure 1H–L*). Quantitative RT-PCR analysis revealed subunit specific expression changes in the first 7 days of life, although only for *eif2b4* were the changes significant (*Figure 1M–Q*).

## Zebrafish *eif2b* subunit mutant alleles generation and molecular characterization

We obtained zebrafish mutant alleles for three *eif2b* subunit genes from the Sanger Institute Zebrafish Mutation Project (*eif2b1*, ENSDARG00000091402; *eif2b2* ENSDARG00000041397; *eif2b4*, ENSDARG00000014004) (*Figure 2A–C*) (15). *eif2b1* (sa12357) harbors a T/A transversion resulting in an early stop in exon 8 (*Figure 2A*); *eif2b2* (sa17223) has a G/A transition in exon 5, mutating an essential splice site (*Figure 2B*); and *eif2b4* (sa17367) has a G/A transition in exon 12 mutating an essential splice site (*Figure 2C*). To mutagenize subunit *eif2b5*, we used the Clustered Regularly Interspaced Short Palindromic Repeats (CRISPR)/Cas9 system and created six different alleles by targeting exon 1 (*Figure 2D*). This created three alleles (*zc101*, *zc103*, and *zc104*) with in-frame deletions leading to loss of amino acids; two alleles with a combination of amino acid loss and missense mutations (*zc105* and *zc106*); and one allele (*zc102*) with a nonsense mutation that caused a frame shift in exon one leading to a premature stop codon in exon 2. Following CRISPR mutagenesis, individual G0 fish were outcrossed to wild-type animals and offspring were screened by high-resolution melt analysis (HRMA) PCR (*Xing et al., 2014*). In addition to HRMA PCR genotyping, we confirmed genotypes by Sanger sequencing both in the G0 and subsequent generations. cDNA sequencing of *eif2b2*$^{sa17223}$, *eif2b5*$^{zc102}$ and *eif2b5*$^{zc103}$ confirmed that the genomic mutations caused splicing failure or the nonsense mutation and frame shift (*Figure 2E*). We tried western blotting with several commercial antibodies against Eif2b2 and Eif2b5 (Eif2B2 Abcam 133848; Eif2B5 Abcam ab91563, GeneTex 30808, Santa Cruz 514056, ProSci 56–847, Bethyl A302-557A) but none gave a specific band.

Most of the alleles in the different subunits were homozygous viable, and survived to adulthood and were fertile. The *eif2b5*$^{zc103/zc103}$ mutants showed a decrease in size compared to their wild type and heterozygous siblings (2.2 cm vs 1.8 cm, 0.3 and 0.2 s.D., and p<0.0001) (*Figure 2F–G*). Mutants of a more severe allele, *eif2b5*$^{zc102/zc102}$, harboring an early stop codon, exhibited growth deficits such as failure to develop a swim bladder, and were smaller in size compared to their wild type and heterozygous siblings (*Figure 2H*, arrowhead). This phenotype was also present in the *eif2b2*$^{sa17223/sa17223}$ allele (*Figure 2H*, arrowhead). Two alleles did not survive past 2 weeks of age: *eif2b5*$^{zc102/zc102}$ and *eif2b2*$^{sa17223/sa17223}$ (at 10 dpf, *eif2b5*$^{+/+}$ WT n = 12; *eif2b5*$^{zc102/+}$ heterozygous n = 26; *eif2b5*$^{zc102/zc102}$ homozygous n = 0. *eif2b2*$^{+/+}$ WT n = 68; *eif2b2*$^{sa17223/+}$ heterozygous n = 118; *eif2b2*$^{sa17223/sa17223}$ homozygous n = 2) (*Figure 2I–J*). By contrast, the milder phenotype *eif2b5*$^{zc103}$ allele survives until adulthood (at 10 dpf, *eif2b5*$^{+/+}$ WT n = 25; *eif2b5*$^{zc103/+}$ heterozygous n = 41; *eif2b5*$^{zc103/zc103}$ homozygous n = 22) (*Figure 2K*).

Since affected VWM patients often have motor impairment, we analyzed the functional effects of *eif2b2*, *eif2b4*, and *eif2b5* mutants on motor behavior from 5 dpf to 7 dpf. The *eif2b2*$^{sa17222sa172232}$ homozygous mutants showed reduction in distance moved, movement time, and velocity at 7dpf (*Figure 2L*). The *eif2b4*$^{sa17367/sa17367}$ mutants also had impairments in growth, motor function, and survival (*Figure 2—figure supplement 1A-C*). The *eif2b5*$^{zc102/zc102}$ mutants had motor function impairments by 5 dpf (*Figure 2L*). On the other hand, the *eif2b5*$^{zc103/zc103}$ mutants had no difference in distance moved, movement time, or velocity as compared to their wild-type siblings (*Figure 2L*). To exclude the possibility that the reduced motor function was due to the developmental delay, we

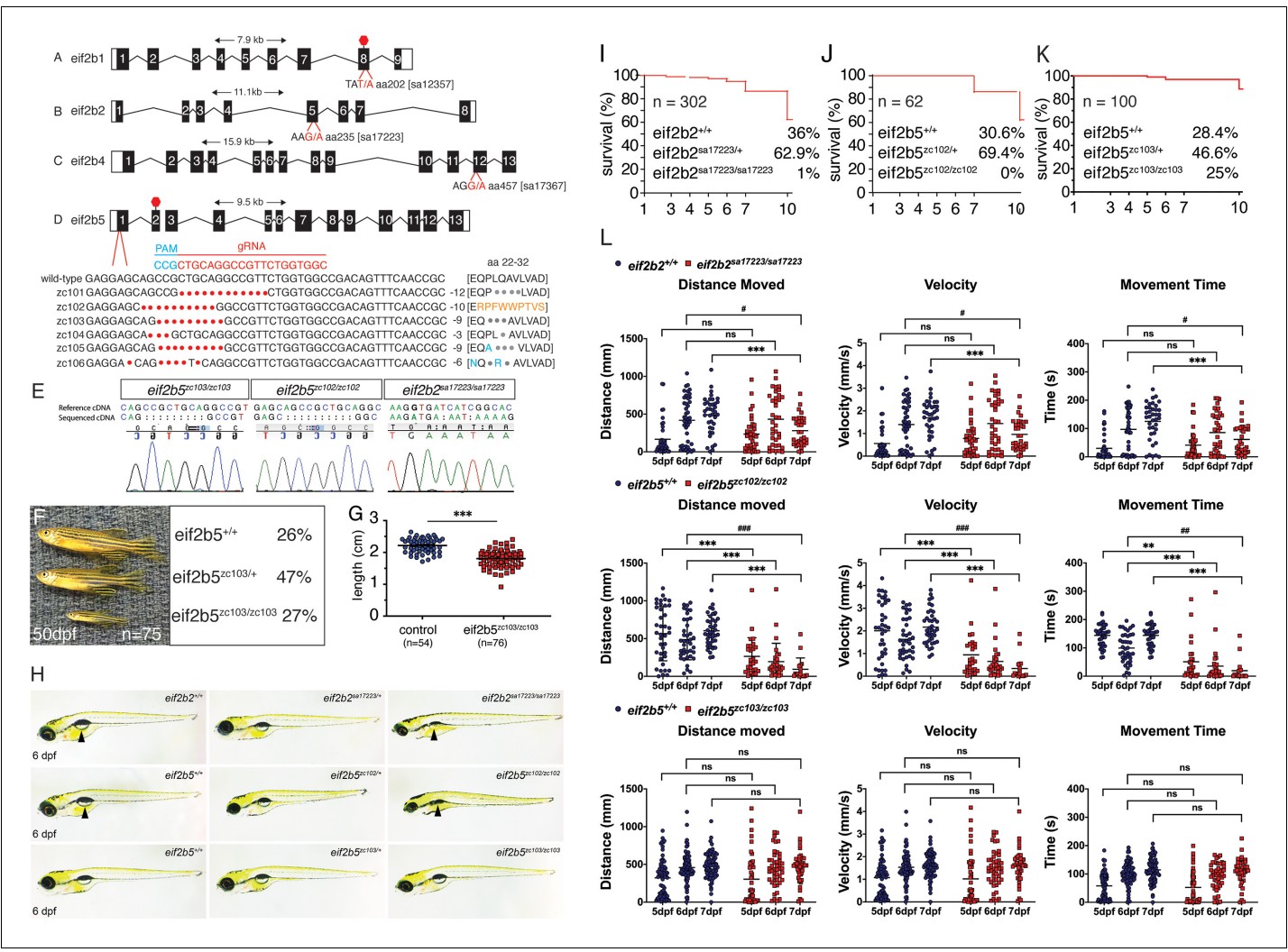

**Figure 2.** Zebrafish *eif2b* allele generation and characterization. (A-D) Depiction of zebrafish *eif2b* subunits exon structure and the location and nucleotide change for each mutant. (A) *eif2b1* harbors a T/A transversion resulting in an early stop in exon 8. (B) *eif2b2* has a G/A transition in exon 5, mutating an essential splice site. (C) *eif2b4* has a G/A transition in exon 12 mutating an essential splice site. (D) *eif2b5* exon one was targeted for mutagenesis using a gRNA (red). Six distinct alleles were recovered (described in text). (E) Chromatograms of cDNA confirm presence of predicted mutations for *eif2b5^{zc103/zc103}*, *eif2b5^{zc102/zc102}*, and *eif2b2^{sa17223/sa17223}*. (F) *eif2b5^{zc103/zc103}* mutants survive until adulthood in Mendelian ratios, but show grow defects compared to their heterozygous and wild-type siblings. (G) Adult *eif2b5^{zc103/zc103}* lengths are significantly shorter compared to their wild-type and heterozygous siblings. (H) Bright-field (BF) images of 6 dpf *eif2b5^{zc103/zc103}*, *eif2b5^{zc102/zc102}*, and *eif2b2^{sa17223/sa17223}* larva. *eif2b2^{sa17223/sa17223}* and *eif2b5^{zc102/zc102}* have no swim bladder (arrowhead) and a small head. (I) Kaplan-Meyer survival curves from an *eif2b2^{sa17223/+}* heterozygous in-cross shows 1% (n = 3) homozygote survival at 10 dpf (total n = 302); however no homozygotes live past 2 weeks of age. (J) Kaplan-Meyer survival curves from an *eif2b5^{zc102/+}* heterozygous in-cross shows that all homozygotes were dead by 10 dpf (total n = 62). (K) Kaplan-Meyer survival curves from an *eif2b5^{zc103/+}* heterozygous in-cross show no mortality of homozygotes. (L) Motor swimming analysis shows impaired swimming behavior in mutants. Distance moved, time spent moving, and velocity, for wild-type controls, and *eif2b5^{zc103/zc103}*, *eif2b5^{zc102/zc102}*, and *eif2b2^{sa17223/sa17223}* mutants, at 5, 6, and 7 dpf. Mean shown with 95% confidence intervals.

The online version of this article includes the following source data and figure supplement(s) for figure 2:

**Source data 1.** Quantification of lengths.
**Source data 2.** Quantification of behavior results.
**Source data 3.** Quantification of behavior results of *eif2b4^{sa17367/sa17367}* allele.
**Source data 4.** qRT-PCR for ISR transcripts for *eif2b4^{sa17367/sa17367}* allele.
**Source data 5.** Survival quantification for *eif2b4^{sa17367/sa17367}* allele.
**Source data 6.** Survival quantification for *eif2b5^{zc103/zc103}* allele.
**Figure supplement 1.** Zebrafish *eif2b4^{sa17367/sa17367}* allele characterization.

compared the behavior of 7 dpf $eif2b2^{sa17223}$ or $eif2b5^{zc102}$ mutants, to 6dpf wild-type siblings, and still found abnormal motor function in mutants (*Figure 2L*). However, since some of the mutants (such as $eif2b5^{zc102}$) had phenotypes including reduced viability, smaller body size, etc., it is possible that some of the behavioral deficits are not due to problems of the CNS. Since the *eif2b2* and *eif2b5* mutations are the most commonly reported in clinical patients (*van der Knaap et al., 2003a*), the $eif2b2^{sa17223}$, $eif2b5^{zc102}$, and $eif2b5^{zc103}$ mutants were used for the remaining experiments in this manuscript. *eif2b* mutants had abnormal CNS development.

Because we observed that the *eif2b* mutants had early lethality and abnormal growth, we evaluated different CNS markers in early development. We compared three *eif2b* alleles: two alleles in two different subunits with early lethality, $eif2b5^{zc102}$ and $eif2b2^{sa17223}$; and a homozygous viable allele, $eif2b5^{zc103}$. At 5 dpf, all three of these mutants showed an increase in cell death, quantified using terminal deoxynucleotidyl transferase dUTP nick-end labeling (TUNEL) ($eif2b5^{zc103}$: mean = 129.1, S.D. = 79.5, $eif2b5^{+/+}$: mean = 12.2, S.D. = 9.8, p=0.001; $eif2b5^{zc102}$: mean = 163.7, S.D. = 78.2 $eif2b5^{+/+}$: mean = 18.7, S.D. = 10.6 $p=2.03\times10^{-5}$; $eif2b2^{sa17223}$: mean = 202.8, S. D. = 60.8: $eif2B2^{+/+}$: mean = 37.4, S.D. = 25.2, p=0.0005) (*Figure 3A–E*). We also compared counts of cells expressing phospho-histone H3 (pH3), an indicator of cell proliferation. $eif2b5^{zc103}$ and $eif2b5^{zc102}$ mutants showed no change in pH3 cell counts at 5 dpf compared to controls, but showed a noticeable change in proliferation pattern: a lack of proliferation in the outer perimeter of the eyes (arrow) and the tectum (arrowhead), but increased proliferation in the ventricular region (asterisk) ($eif2b5^{zc103}$: mean = 247.2, S.D. = 44.5, $eif2b5^{+/+}$: mean = 273.1, S.D. = 43.8, p=0.206; $eif2b5^{zc102}$: mean = 243.3, S.D. = 40.0, $eif2b5^{+/+}$: mean = 288.3, S.D. = 32.0, p=0.0238) (*Figure 3F–H,J*). $eif2b2^{sa17223}$ mutants showed a decrease in pH3 cell counts at 5 dpf compared to controls as well as a change in proliferation pattern ($eif2b2^{sa17223}$: mean = 185.3, S.D. = 37.8: $eif2b2^{+/+}$: mean = 271.6, S.D. = 46.7, p=0.0004) (*Figure 3I,J*).

In the spinal cord, we measured counts of oligodendrocyte precursor cells (OPCs), which give rise to myelin-producing oligodendroglia. We used the Tg(*olig2:dsRed*) line crossed into the $eif2b5^{zc103/zc103}$, $eif2b5^{zc102/zc102}$ or $eif2b2^{sa17223/sa172233}$ background (*Kucenas et al., 2008*). The Tg(*olig2:dsRed*) line labels OPCs as well as motor neurons and neuronal precursors, but in the spinal cord is specific for OPCs in labeled cells that have migrated dorsally. We observed no change in the number of OPCs (measured in two somites directly above the caudal end of the yolk) ($eif2b5^{zc103}$: mean = 26.6, S.D. = 7.55, $eif2b5^{+/+}$: mean = 30.71, S.D.=,5.88 p=0.278; $eif2b5^{zc102}$: mean = 30.6, S. D. = 11.0, $eif2b5^{+/+}$: mean = 36.6, S.D. = 9.63, p=0.514; $eif2b2^{sa17223}$: mean = 35.8, S.D. = 5.36, $eif2b2^{+/+}$: mean = 39.2, S.D. = 4.66, p=0.163) (*Figure 3K–O*). We evaluated apoptotic OPCs in the brain by crossing the Tg(*olig2:dsRed*) line into the $eif2b5^{zc103/zc103}$, $eif2b5^{zc102/zc102}$ or the $eif2b2^{sa17223/sa172233}$ background and staining with TUNEL. All three lines showed significant increases in the number of TUNEL-positive OPCs specifically in the hindbrain ($eif2b5^{zc102}$: mean = 17.5, S.D. = 3.02, $eif2b5^{+/+}$: mean = 1, S.D. = 1, p=0.00004; $eif2b5^{zc103}$: mean = 3, S. D. = 2.26, $eif2b5^{+/+}$: mean = 0.7, S.D. = 1.06, p=0.0144; $eif2b2^{sa17223}$: mean = 3.5, S.D. = 1.05, $eif2b2^{+/+}$: mean = 0.333, S.D. = 0.577, p=0.002) (*Figure 3P–T*).

To further quantify effects on oligodendrocytes, we analyzed the expression of *myelin associated glycoprotein* in differentiated oligodendrocytes by in situ hybridization, and showed no difference in numbers of those cells in the spinal cord of mutants compared with wild-type siblings (*Figure 4*). To confirm that the increased apoptosis was due to loss of OPCs, we counterstained for the OPC marker, *olig1* along with the TUNEL assay in the $eif2b5^{zc103/zc103}$, $eif2b5^{zc102/zc102}$ or $eif2b2^{sa17223/sa172233}$ mutants (*Figure 4*).

The data above showed that proliferation patterns were altered in *eif2b* mutants, and in order to discover in which cell types this occurred, we counterstained with the radial glia marker (Zrf1), the pan-neuronal maker (HuC/D), and the microglial marker (*apoeb*) along with phospho-histone3 (pH3). The Zrf1 and pH3 double-positive cells showed no difference in either the tectum or the eye in *eif2b* mutants. In contrast, the midline of $^{eif2b5zc102/zc102}$ contained significantly more Zrf1 and pH3 double-positive cells, while $eif2b2^{sa17223/sa17223}$ and $eif2b5^{zc103/zc103}$ tended towards increased numbers (*Figure 5A,B*). The HuC/D and pH3 double-positive cells were significantly decreased in the tectum and tended towards decreased numbers in the eye of $eif2b5^{zc102/zc102}$ and $eif2b2^{sa17223/sa17223}$ mutants. But there were no differences in either the tectum or eye of $eif2b5^{zc103/zc103}$ compared to wild-type siblings (*Figure 5A,B*). The *apoeb* and pH3 double-positive cells were increased in the midline of *eif2b5* mutants but not in the tectum and eye (*Figure 5C,D*). On the other hand, the

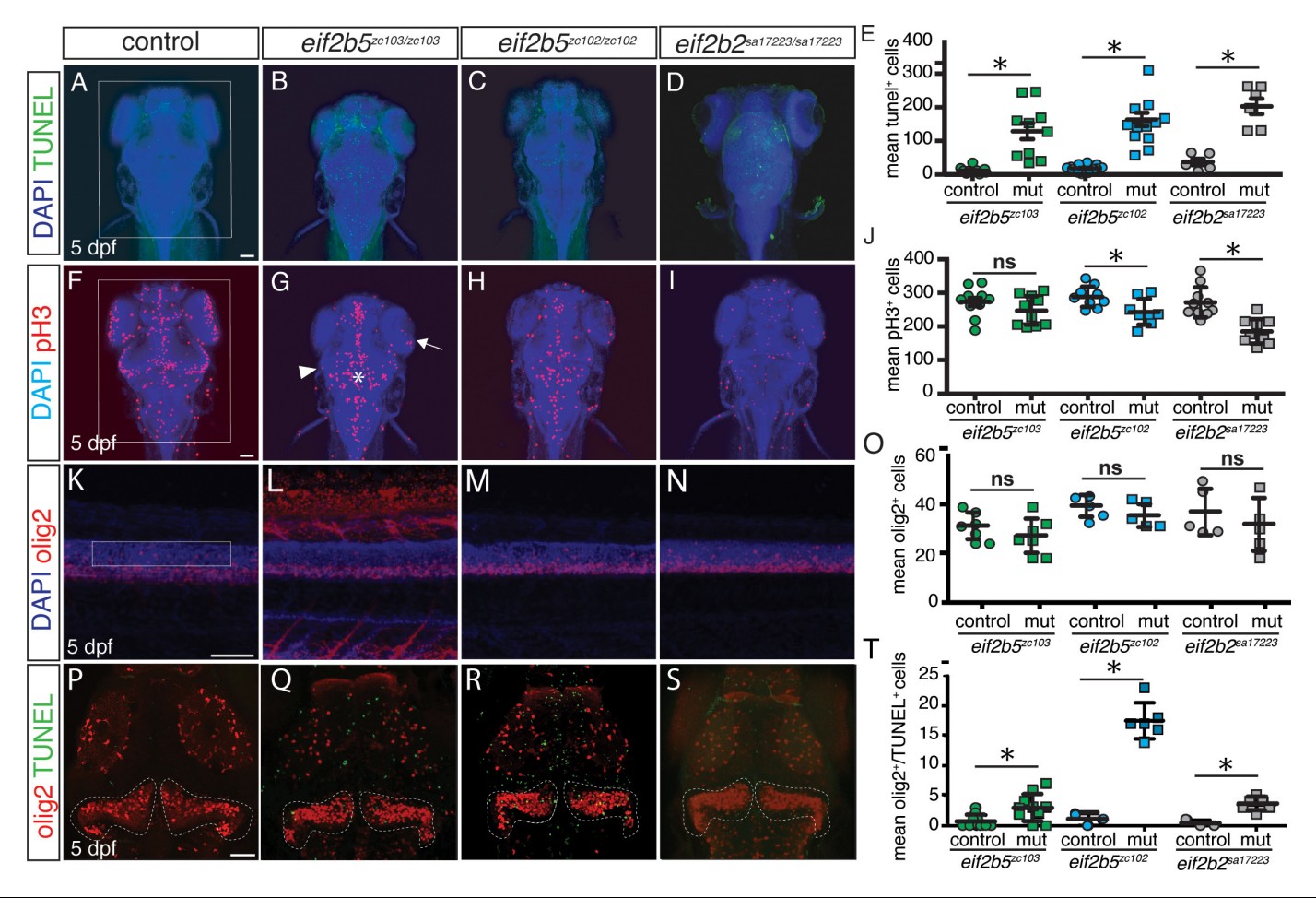

**Figure 3.** *eif2b* mutants demonstrate abnormal CNS development. Confocal images, z-stack maximal projections. (**A–I**), dorsal views of the brain, rostral to the top, scale bar 50 µm; (**K–N**), lateral views of spinal cord, dorsal to the top, scale bar 50 um. (**P–S**), dorsal views of brain, rostral to top, scale bar 50 µm. *p<0.05. (**A–D**) TUNEL and DAPI staining shows increased apoptosis in homozygous mutant alleles compared to controls (wild-type and heterozygous siblings) in eif2b5$^{zc103/zc103}$ eif2b5$^{zc102/zc102}$ and *eif2b2*$^{sa17223/sa17223}$ mutants. (**E**) Quantification of mean TUNEL+ cell counts in eif2b5$^{zc103/zc103}$ eif2b5$^{zc102/zc102}$ and *eif2b2*$^{sa17223/sa17223}$ mutants compared to sibling controls. (**F–I**) Phospho-histone 3 and DAPI staining shows decreased cell proliferation in 5 dpf *eif2b2*$^{sa17223/sa17223}$ mutants compared to controls, while eif2b5$^{zc103/zc103}$ eif2b5$^{zc102/zc102}$ mutants show a change in proliferation pattern, specifically in the optic tectum. (**J**) Quantification of mean number pH3+ cells counts in eif2b5$^{zc103/zc103}$ eif2b5$^{zc102/zc102}$ and *eif2b2*$^{sa17223/sa17223}$ mutants compared to sibling controls. (**K–N**) Olig2dsRed and DAPI staining shows no change in OPC counts in the spinal cords of 5 dpf eif2b5$^{zc103/zc103}$ eif2b5$^{zc102/zc102}$ and *eif2b2*$^{sa17223/sa17223}$ mutants compared to controls. (**O**) Quantification of mean number Olig2dsRed+ counts in eif2b5$^{zc103/zc103}$ eif2b5$^{zc102/zc102}$ or *eif2b2*$^{sa17223/sa17223}$ mutants compared to sibling controls. (**P–S**) Co-labeled Olig2dsRed+/TUNEL+ cell counts staining shows increase in Olig2dsRed+ cells undergoing apoptosis in brains of 5 dpf eif2b5$^{zc103/zc103}$, eif2b5$^{zc102/zc102}$, or *eif2b2*$^{sa17223/sa17223}$ mutants compared to sibling controls. (**O**) Quantification of mean number of co-labeled Olig2dsRed+/TUNEL+ cell counts in eif2b5$^{zc103/zc103}$, eif2b5$^{zc102/zc102}$, or *eif2b2*$^{sa17223/sa17223}$ mutants compared to sibling controls.

The online version of this article includes the following source data for figure 3:

**Source data 1.** Quantification of TUNEL, pH3, olig2, and olig2/TUNEL results.
**Source data 2.** Quantification of TUNEL, pH3, olig2, and olig2/TUNEL results.
**Source data 3.** Quantification of TUNEL, pH3, olig2, and olig2/TUNEL results.
**Source data 4.** Quantification of TUNEL, pH3, olig2, and olig2/TUNEL results.

*eif2b2*$^{sa17223/sa17223}$ mutation had no impact on the *apoeb* and pH3 double-positive cells in the tectum, the midline, or the eye (***Figure 5C,D***). Thus, interestingly, the disordered proliferation was a widespread phenotype in different cell lineages, but did also show spatial specificity.

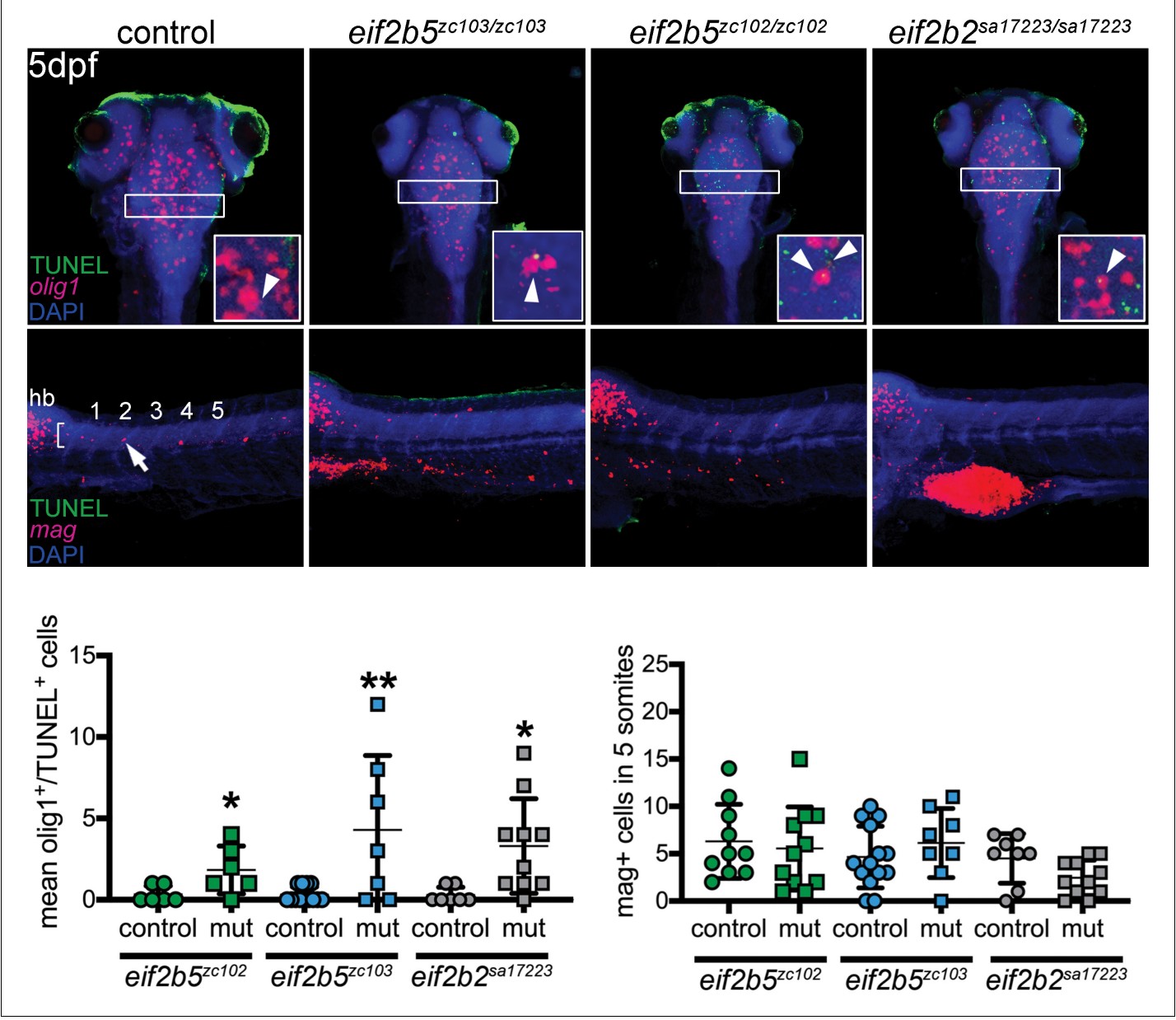

**Figure 4.** Top row (and quantification in lower left): increased apoptotic OPCs in the brain and spinal cord. Confocal images of brain, z-stack, rostral to the top, double-labeling for TUNEL and *olig1* (in situ probe), in WT, *eif2b5*<sup>zc103/zc103</sup>, *eif2b5*<sup>zc102/zc102</sup> or *eif2b2*<sup>sa17223/sa172233</sup> larvae. Region used for quantification shown by box. Inset in each panel shows example of double-labeled cell (TUNEL, *olig1*) for each genotype (except WT), single confocal slice image. Middle panels: no change in apoptosis of differentiated oligodendrocytes, co-labeled with *myelin associated glycoprotein* (*mag*) (in situ probe) and TUNEL. Confocal z-stack images of spinal cord, rostral to the left; quantified in lower right panel.
The online version of this article includes the following source data and figure supplement(s) for figure 4:

**Source data 1.** Quantification of *olig1*, TUNEL and *mag* results.
**Figure supplement 1.** Localization of *mag* in situ probe in oligodendrocytes; double-labeling for *mag* and GFP, in Tg(*sox10:GFP*) line.

In order to identify the cell type of apoptotic cells outside the hindbrain, we stained with Zrf1, HuC/D, and *apoeb,* and counterstained with TUNEL. This revealed that only *eif2b5*<sup>zc102/c102</sup> mutants had a significant increase in Zrf1 and TUNEL double-positive cells in the midline (*Figure 5*).

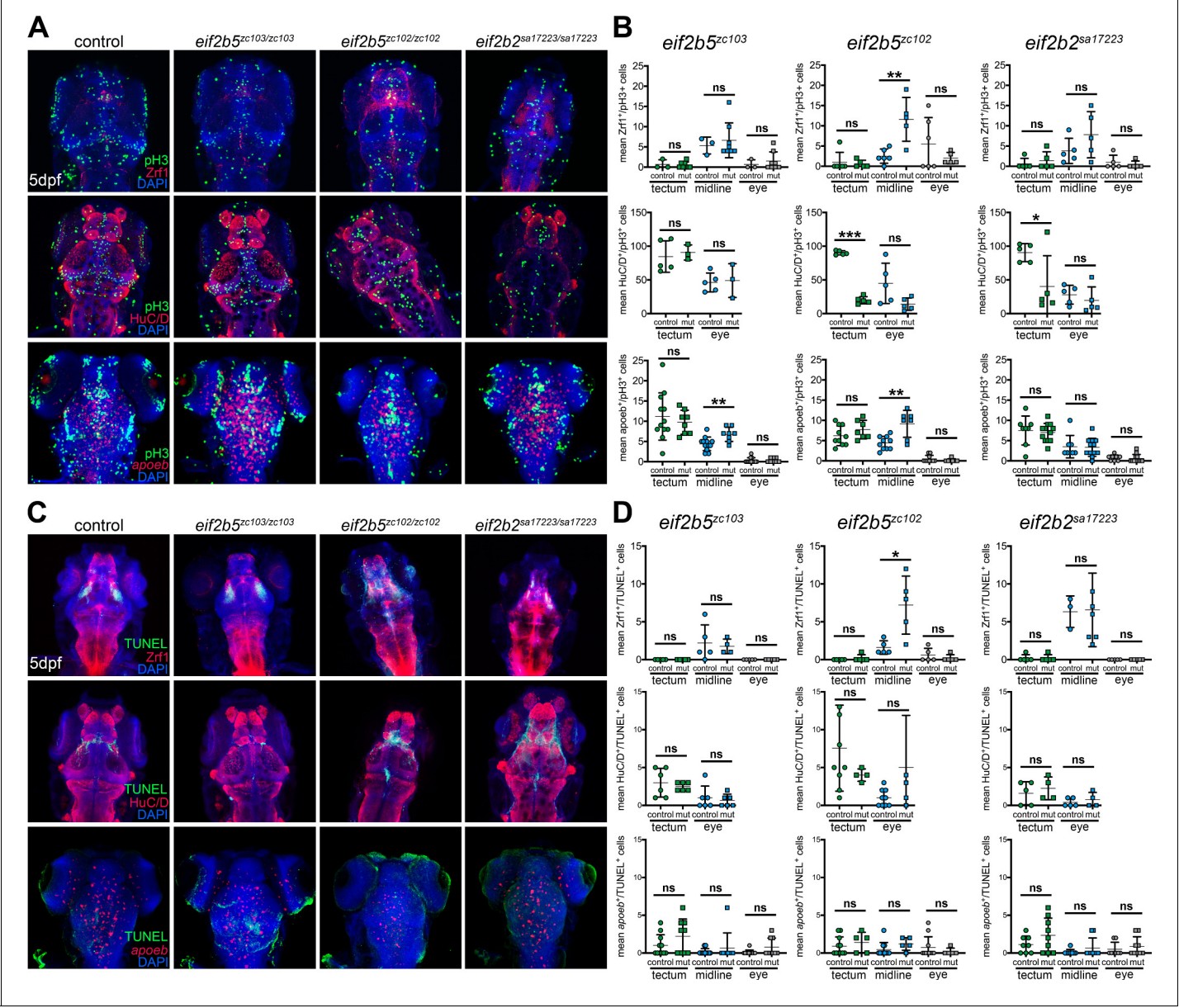

**Figure 5.** Determination of cell types affected in mutants by changes in proliferation or in apoptosis. (A) Confocal images of brain, z-stack, rostral to the top, in WT, *eif2b5*<sup>zc103/zc103</sup>, *eif2b5*<sup>zc102/zc102</sup> or *eif2b2*<sup>sa17223/sa172233</sup> larvae. Confocal images of brain, z-stack, rostral to the top. Top row, labeling for Zrf1, DAPI, and pH3. Middle row, labeling for HuC/D, DAPI, and pH3. Bottom row, labeling for *apoeb*, DAPI, and pH3. (B) Quantification of cell counts in (A). (C) Confocal images of brain, z-stack, rostral to the top, in WT, *eif2b5*<sup>zc103/zc103</sup>, *eif2b5*<sup>zc102/zc102</sup> or *eif2b2*<sup>sa17223/sa172233</sup> larvae. Confocal images of brain, z-stack, rostral to the top. Top row, labeling for Zrf1, DAPI, and TUNEL. Middle row, labeling for HuC/D, DAPI, and TUNEL. Bottom row, labeling for *apoeb*, DAPI, and TUNEL. (D) Quantification of cell counts in (C).

The online version of this article includes the following source data for figure 5:

**Source data 1.** Quantification of Zrf1, HuC/D, and *apoeb*, double-labeling with pH3, results in tectum, midline, and eyes in *eif2b5*<sup>zc103/zc103</sup>, *eif2b2*<sup>zc102/zc102</sup>, and *eif2b2*<sup>sa17223/sa172233</sup>.

**Source data 2.** Quantification of Zrf1, HuC/D, and *apoeb*, double-labeling with TUNEL, results in tectum, midline, and eyes in *eif2b5*<sup>zc103/zc103</sup>, *eif2b2*<sup>zc102/zc102</sup>, and *eif2b2*<sup>sa17223/sa172233</sup>.

## Zebrafish adult *eif2b5$^{zc103/zc103}$* mutants show myelin defects

VWM patients have a loss of white matter and abnormal white matter signal quality on MRI (*Leegwater et al., 2003*; *van der Knaap et al., 2006*). We used transmission electron microscopy (TEM) to measure myelin content in the adult *eif2b5$^{zc103/zc103}$* mutants compared to age-matched controls. The adult mutants showed a decrease in the thickness of the myelin sheath as measured by estimation of the G-ratio, the perimeter of the axon to the perimeter of the myelin sheath (*eif2b5$^{zc103}$*: mean = 0.690, S.D. = 0.076, *eif2b5$^{+/+}$*: mean = 0.593, S.D. = 0.085, p=2.18E-22) (*Figure 6A–E*). We also performed Black Gold II staining to compare myelination patterns in the CNS. The adult mutants showed a disorganized pattern of myelinated axons in the optic tectum compared to age-matched wild-type adult controls (*Figure 6F,G*).

We performed magnetic resonance imaging (MRI) of adult *eif2b5* mutants, and found a decrease in overall brain size in the adult *eif2b5$^{zc103/zc103}$* mutants compared to adult wild-type controls using the skull as a normalization factor to account for the decrease in overall size of the mutants (*Figure 6H–J*; *Figure 6—figure supplement 1*). These images showed a decrease in overall brain volume of the *eif2b5$^{zc103/zc103}$* mutants as measured by length, width and height of the brains compared to age-matched controls (*eif2b5$^{zc103/zc103}$* normalized length: mean = 0.933; S.D. = 0.041; *eif2b5$^{+/+}$* normalized length: mean = 1.11, S.D. = 0.091, p=0.001; *eif2b5$^{zc103/zc103}$* normalized width: mean = 0.836; S.D. = 0.034; *eif2b5$^{+/+}$* normalized width: mean = 0.958, S.D. = 0.023, p=0.00002; *eif2b5$^{zc103/zc103}$* normalized height: mean = 0.551; S.D. = 0.018; *eif2b5$^{+/+}$* normalized height: mean = 0.629, S.D. = 0.035, p=0.0006) (*Figure 6—figure supplement 1*). We also found a change in the intensity between the grey of the optic tectum and the white matter of the periventricular grey zone in the T2 MRI images from the rhombencephalic ventricle (RV) at the end of the midbrain moving rostrally (*eif2b5$^{zc103/zc103}$* normalized intensity: mean = 0.849; S.D. = 0.108; *eif2b5$^{+/+}$* normalized length: mean = 0.743, S.D. = 0.091, p=0.0005) (*Figure 6J*).

## Human *EIF2B* genes rescue zebrafish *eif2b* mutant lethality and CNS apoptosis

Although the human and zebrafish *eif2b* subunit genes show significant conservation in protein sequence, an important question for modeling VWM is to demonstrate functional conservation. Zebrafish and human *eif2b2* ortholog amino acids are highly conserved, and both orthologs have the same number of exons (*Figure 7A–C*). To test functional conservation, we created transgenic animals ubiquitously-expressing human *EIF2B2*. We created a Tol2 transposable vector in which the human *EIF2B2* gene and enhanced green fluorescent protein (eGFP) was expressed under control of the ubiquitously-expressing promoter ß-actin (*Figure 7D*). We mated *eif2b2$^{sa17223/+}$* heterozygous adults, and injected their embryos at the one-cell stage with the human *EIF2B2* vector to introduce the transgene (Tg) into the germline. Embryos were screened at 3 dpf, and those positive for GFP were grown up to adults and genotyped to identify and generate stable lines (*Figure 7F*). The transgenic mutant *eif2b2$^{sa17223/\ sa17223}$*;Tg(ß-actin:EIF2B2-2A-eGFP) larva developed a swim bladder and the growth deficits were rescued (*Figure 7F*, arrowhead). As adults, the GFP positive *eif2b2$^{sa17223/+}$*; Tg(ß-actin:EIF2B2-2A-eGFP) were crossed with non-transgenic *eif2b2$^{sa17223/+}$*. The offspring from this cross were genotyped and compared to a non-transgenic *eif2b2$^{sa17223/+}$* in-cross. Typically, *eif2b2$^{sa17223/\ sa17223}$* have early lethality by 10 dpf. In contrast the transgenic homozygous mutant *eif2b2$^{sa17223/\ sa17223}$*;Tg(ß-actin:EIF2B2-2A-eGFP) animals showed 21% survival at 10 dpf (e.g. Mendelian ratios) (*Figure 7E*). This increase in transgenic homozygous mutant survival continued to adulthood, and the mutants were able to mate and were fertile. Further, the *eif2b2$^{sa17223/sa17223}$* increase in apoptosis (*Figure 7G–H,K*) was rescued and apoptosis in the *eif2b2$^{sa17223/sa17223}$*;Tg(ß-actin:EIF2B2-2A-eGFP) was significantly reduced (TUNEL counts *eif2b2$^{sa17223/sa17223}$* transgenic: mean = 31.5, S.D. = 12.9; *eif2b2$^{sa17223/sa17223}$* non-transgenic: mean = 84.4, S.D. = 27.63, p=1.89E-05) (*Figure 7I–K*). *eif2b* mutants express a truncated *eif2b5* transcript that causes ISR activation and defects in behavior.

Work studying cancer cell lines has shown that during periods of hypoxia, ISR activation leads to retention of intron 12 in *EIF2B5* by interfering with the exon 12 splice site (*Brady et al., 2017*; *Figure 8A*). This retained intron contains a premature stop codon, resulting in a truncated 65 kDa EIF2B5 protein, and rendering the eIF2B complex unable to initiate translation. To test whether the ISR in VWM could similarly affect *eif2b5* splicing, we tested intron 12 retention. We found an

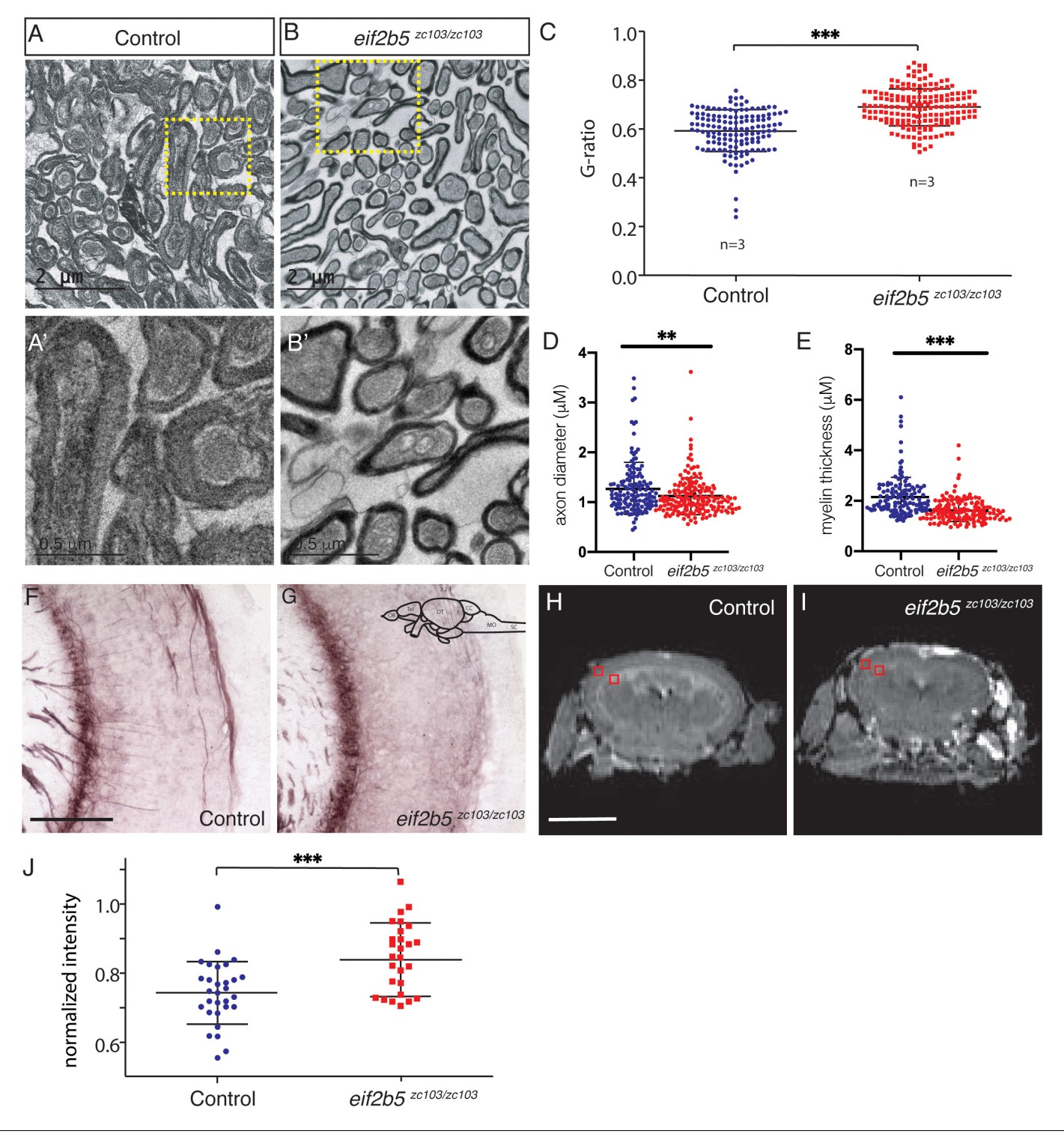

**Figure 6.** Adult ei*f2b5* ^zc103/zc103^ mutants show myelin defects. (**A–C**) Transmission electron microscopy (TEM) of adult ei*f2b5*^zc103/zc103^ optic nerve compared to ei*f2b5*^+/+^ adults. Scale bar 2 µm. (**A**) ei*f2b5*^+/+^ adult TEM image. (**B**) ei*f2b5*^zc103/zc103^ adult TEM image. (**A', B'**) Higher magnification views of insets represented in (**A**) and (**B**), scale bar 0.5 µm. (**C–E**). Comparison of ratio between axon perimeter and myelin sheath perimeter, G-ratio, between ei*f2b5*^zc103/zc103^ and ei*f2b5*^+/+^; axon diameter and myelin thickness quantification shown. (**F–G**) Black Gold stain of adult ei*f2b5*^zc103/zc103^ optic tectum compared to ei*f2b5*^+/+^ adults. Scale bar 5 µm. (**F**) ei*f2b5*^+/+^ adult Black Gold stained image. (**G**) ei*f2b5*^zc103/zc103^ adult Black Gold stained image. (**H–J**) Magnetic Resonance Image (MRI) of adult ei*f2b5*^zc103/zc103^ compared to ei*f2b5*^+/+^. Scale bar 2 mm. Slice scheme of MRI images from the rhombencephalic ventricle (RV) at the end of the midbrain moving rostrally. (**H**) ei*f2b5*^+/+^ adult T2 MRI image slice 1. (**I**) ei*f2b5*^zc103/zc103^ adult T2 MRI of

*Figure 6 continued on next page*

*Figure 6 continued*

slice 1. (J) T2 intensity analysis. The normalized change in intensity from the from grey matter region of the optic tectum to the white matter region of the periventricular grey zone of the optic tectum, indicated by red boxes.

The online version of this article includes the following source data and figure supplement(s) for figure 6:

**Source data 1.** Quantification of TEM results.

**Figure supplement 1.** MRI images of adult wild-type control siblings and *eif2b5^zc103/zc103* fish, showing decreased head and body size.

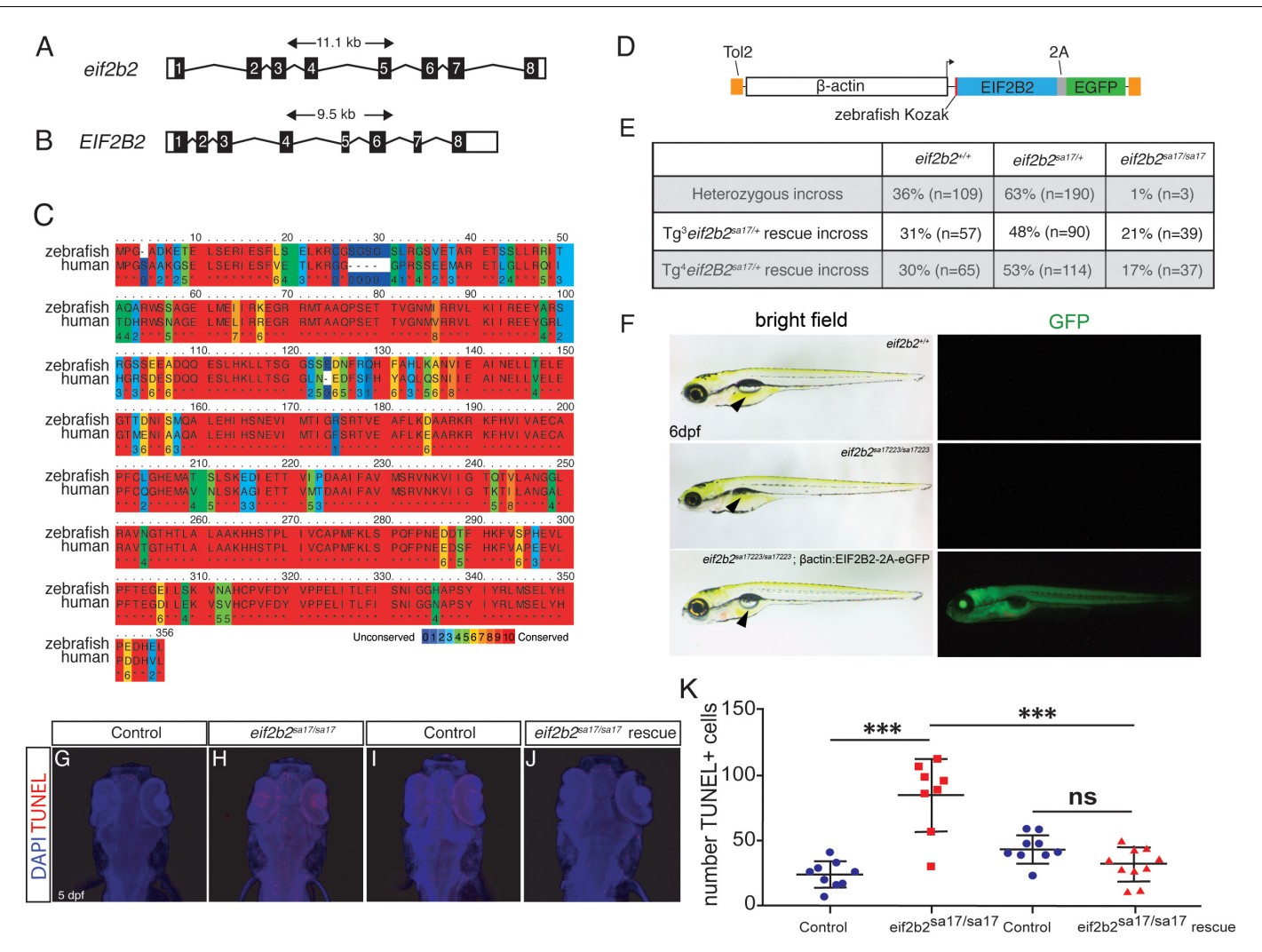

**Figure 7.** Human *EIF2B2* gene rescues zebrafish *eif2b2^sa17223/sa17223* mutants. (A) 11.1 kb zebrafish *eif2b2* gene structure. (B) 9.5 kb human *EIF2B2* gene structure. (C) Conservation of amino acid sequence between zebrafish *eif2b2* and human *EIF2B2*. (D) Schematic of rescue construct containing Tol2, β-actin, human *EIF2B2*, and eGFP. (E) Genotype results (at adult age), of an *eif2b2^sa17223/+* heterozygous incross, and an *eif2b2^sa17223/+* heterozygous zebrafish crossed with two different transgenic alleles (#3 and #4): *eif2b2^sa17223/+*;Tg^3(*β-actin:EIF2B2:2A:eGFP*) or *eif2b2^sa17223/+*;Tg^4(*β-actin:EIF2B2:2A: eGFP*) heterozygous zebrafish. (F) Bright-field and immunofluorescent images of *eif2b2^sa17223/sa17223*;*β-actin:EIF2B2:2A:eGFP* mutant fish: showing a swim bladder and regular-sized head; or showing GFP expression in *eif2b2^sa17223/sa17223*;*β-actin:EIF2B2:2A:eGFP* mutant fish. (G–J) TUNEL and DAPI antibody staining of wild-type control; *eif2b2^sa17223/sa17223* mutant; *eif2b2^+/+*;*β-actin:EIF2B2:2A:eGFP* wild-type control; and *eif2b2^sa17223/sa17223*;*β-actin: EIF2B2:2A:eGFP* mutant. (K) Quantification of TUNEL+ cells.

The online version of this article includes the following source data for figure 7:

**Source data 1.** Quantification of TUNEL results.

**Source data 2.** Quantification of behavior results.

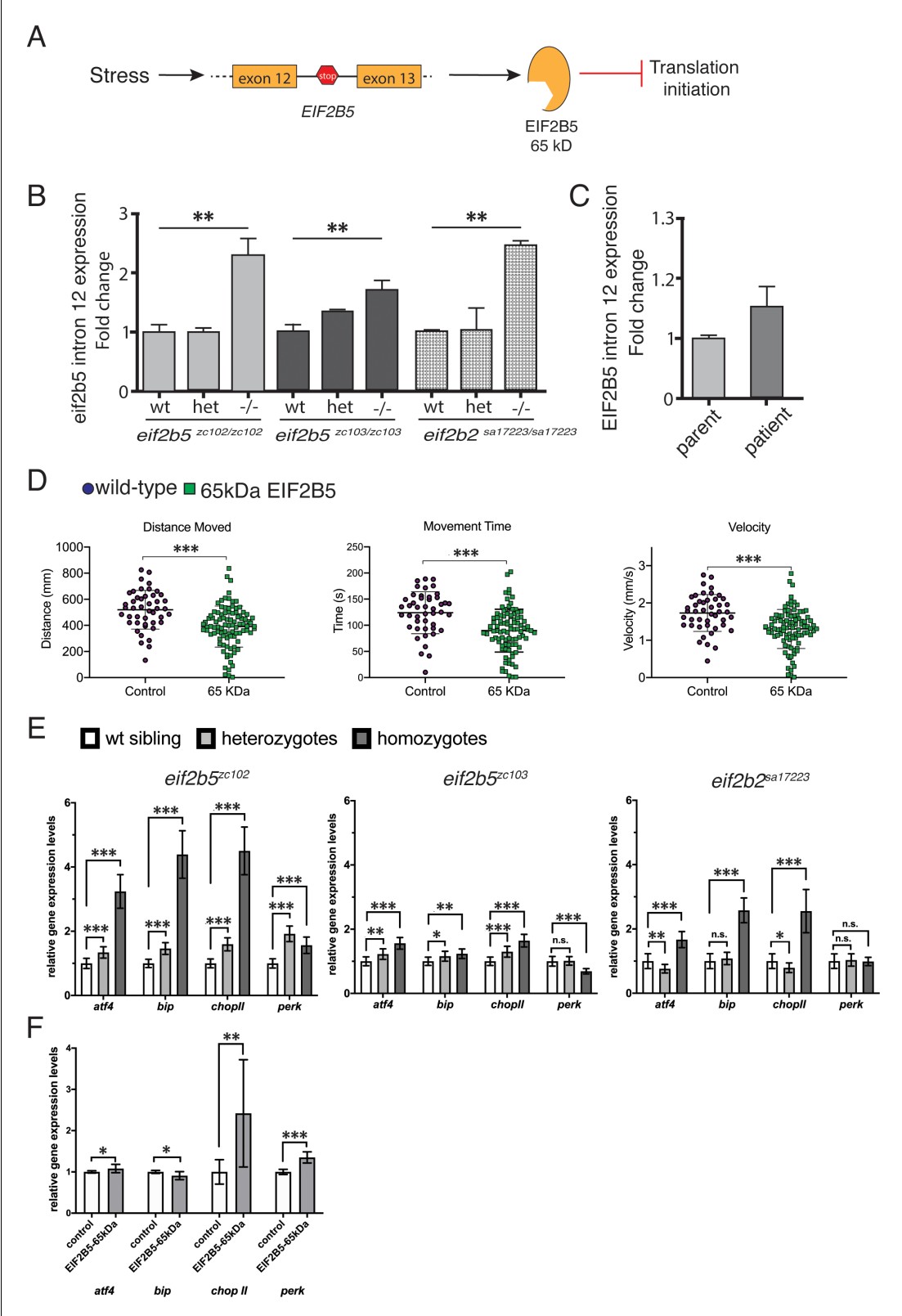

**Figure 8.** Zebrafish *eif2b5* mutant alleles show intron retention, activated ISR, and role of truncated Eif2b5 in worsening phenotype. (**A**) Schematic showing integrated stress response activated intron 12 retention of *eif2b5* resulting in premature stop codon and truncated form of EIF2B5. (**B**) Fold change of *eif2b5* intron 12 expression with qRT-PCR in *eif2b5*<sup>zc103/zc103</sup>, *eif2b5*<sup>zc102/zc102</sup>, and *eif2b2*<sup>sa17223/sa17223</sup> mutants relative to controls. (**C**) qRT-PCR for intron 12 expression in a VWM patient and their control unaffected father. (**D**) Control-injected larvae, compared to those injected with

*Figure 8 continued on next page*

*Figure 8 continued*

truncated *eif2b5* construct, shows impaired swimming behavior; distance moved, time spent moving, and velocity, at 5 dpf. (E) qRT-PCR for *atf4*, *bip*, *chopII*, and *perk* ISR transcripts shows increased expression in eif2b5*zc103/zc103*, eif2b5*zc102/zc102*, and eif2b2*sa17223/sa17223* mutants. (F) qRT-PCR for ISR transcripts (*atf 4*, *bip*, *chop II*, and *perk*) shows increased expression following injection of truncated *eif2b5* construct.

The online version of this article includes the following source data for figure 8:

**Source data 1.** qRT-PCR for intron 12 expression changes (*eif2b5*, zebrafish).
**Source data 2.** qRT-PCR for intron 12 expression changes (*eif2b2*, zebrafish).
**Source data 3.** qRT-PCR for intron 12 expression changes (human).
**Source data 4.** Behavior data for truncated *eif2b5* effects.
**Source data 5.** qRT-PCR for ISR transcript expression.
**Source data 6.** qRT-PCR for ISR transcript expression changes following injection with truncated *eif2b5*.

increase in retention of intron 12 associated with truncated *eif2b5* in *eif2b* mutant alleles (*eif2b5: zc102* and *zc103*; *eif2b2: sa17223*) (*Figure 8B*); and which was also observed in peripheral lymphocytes from a human VWM patient (*Figure 8C*).

To test the effects of the 65 kDa human EIF2B5 on behavior, we expressed truncated EIF2B5 in wild-type embryos. We injected one-cell stage WT embryos with transcribed RNA and evaluated behavior at 5 dpf. We observed impairment of motor behavior (*Figure 8D*), suggesting a dominant-negative effect, although we could not exclude that the effect was caused by over-expression. Since in VWM chronic activation of the ISR is believed to contribute to pathophysiology, we examined ISR activation in *eif2b5* mutants by analyzing the expression of *atf4*, *bip*, *chop II*, and *perk*, which are critical regulators of the ISR. qRT-PCR analysis showed *atf4*, *bip*, and *chop II* expression were increased, but *perk* had variable expression in *eif2b2* and *eif2b5* homozygous mutants (*Figure 8E*), indicating that ISR activation is mainly triggered by *atf4* up-regulation the in *eif2b* mutants. Next, we also examined ISR activation in wild-type embryos expressing truncated EIF2B5. qRT-PCR showed that ISR markers including *atf4*, *chop II*, *bip* and *perk*, were up-regulated (*Figure 8F*), indicating that truncated EIF2B5 was sufficient to activate the ISR.

## Discussion

We have generated and characterized a zebrafish model for VWM by analysis of alleles in *eif2b1*, *2*, *4*; and an allelic series in *eif2b5*. The zebrafish *eif2b* mutants phenocopy important aspects of human VWM, including increased morbidity and mortality, altered myelination, and impaired motor behavior. Further, using rescue with expression of the human EIF2B2 cDNA, we demonstrate functional conservation in the zebrafish *eif2b2* mutant. The mutations we have generated are not specifically homologous to known human VWM patient disease alleles. For therapeutic tests and similar studies future work will benefit from having zebrafish mutants that are homologous to known disease alleles. However, the range of phenotypes in our zebrafish mutants, from lethality in the first several weeks of life, to viability and fecundity, provide an opportunity to further study genotype-phenotype relationships in VWM, which have been noted in patients (*van der Lei et al., 2010*).

Importantly, our data reveals a novel potential mechanism of VWM pathophysiology: ISR activation in the *eif2b* mutants causes abnormal splicing and generation of a truncated *eif2b5* transcript, which is capable of further activating the ISR and impairing motor behavior. Thus, in healthy individuals, stress would activate the ISR but would appropriately be terminated through down-regulation of the ISR by dephosphorylation of eIF2 (*Pakos-Zebrucka et al., 2016*). But in VWM patients, stress would activate the ISR, and would also lead to expression of truncated EIF2B5 expression; which would cause a feed-forward mechanism of further and chronic ISR activation (*Figure 9*). This chronic activation of the ISR is observed in human autopsy samples and in a mouse VWM model (*van der Voorn et al., 2005*; *van Kollenburg et al., 2006*; *Wong et al., 2019*).

The eIF2B complex is comprised of five subunits that together act as the guanine nucleotide exchange factor for eIF2, governing the rate of global protein synthesis and cap-dependent translation initiation. VWM can be caused by autosomal recessive mutations in any of the five subunits of eIF2B. Zebrafish show expression of the *eif2b* genes at 24 hpf through 7 dpf including in the CNS. Mutations in the most commonly affected patient subunits, *eif2b2* and *eif2b5*, showed growth

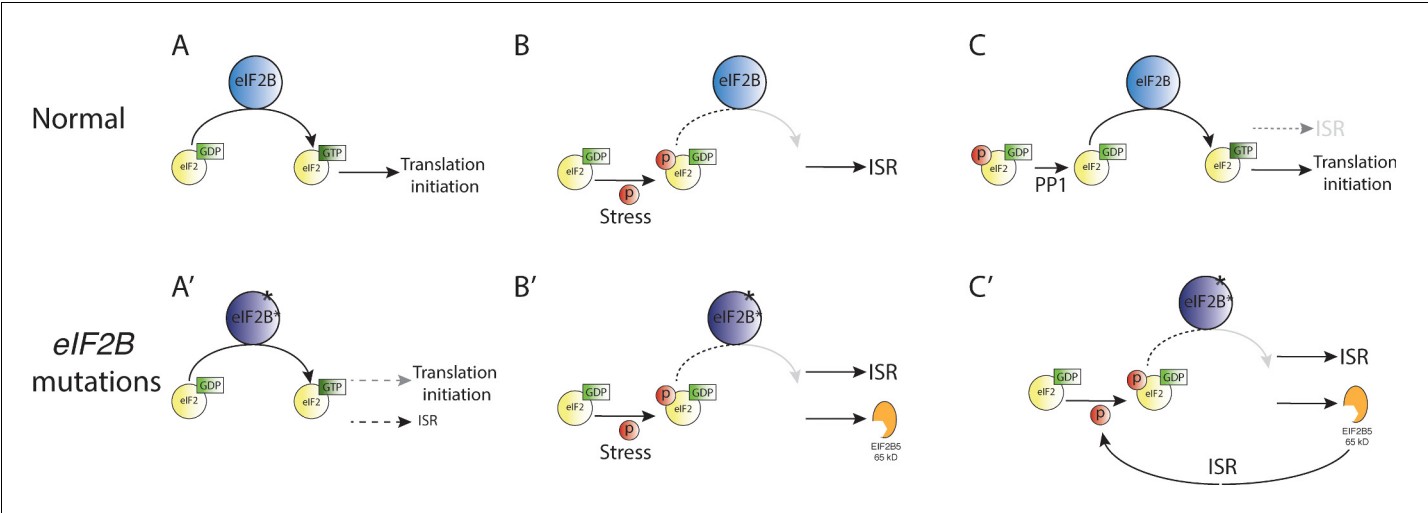

**Figure 9.** Feed-forward effect of truncated EIF2B5. (A–C) Under normal conditions, stress leads to activation of the ISR, which is terminated by dephosphorylation of eIF2a by PP1 (protein phosphatase one complex) and resumption of normal cap-dependent translation. (A'–C') In the presence of *eif2b* subunit mutations in VWM, normal translation is impaired, and there is activation of the ISR above baseline (A'). In response to stress, there is ISR activation and expression of truncated EIF2B5, which leads to chronic activation of the ISR and an inability to terminate the ISR.

deficits and decreased survival, mimicking clinical features of VWM. Interestingly, we found that human EIF2B2 expression in zebrafish can rescue the increased apoptosis and early lethality phenotype of *eif2b2* $^{sa17223}$ homozygous mutants, but not human EIF2B5. The difference between the EIF2B2 and EIF2B5 rescues might be due to the fact that EIF2B5 is the key catalytic subunit of the eIF2B complex for the eIF2•GDP to eIF2•GTP transition, which enhances protein translation. In contrast, expression of human EIF2B5 in zebrafish was lethal, consistent with data showing that stoichiometry of the EIF2B5 subunit (ε) expression is critical (*Wortham et al., 2016*). Taken together, our work suggests conservation of *eif2b* expression and function.

Mutations in the zebrafish eIF2B complex were associated with multiple defects in development of the CNS. In the 5 dpf homozygous mutants of *eif2b2$^{sa17223}$*, *eif2b5$^{zc102}$*, and *eif2b5$^{zc103}$* alleles, there was an increase in apoptosis in the CNS. We also observed altered patterns of cell proliferation pattern in the CNS, and a decrease in overall CNS proliferation in the *eif2b2$^{sa17223}$* mutants. Although there was no change in total OPC numbers in the CNS, we found an increase in the number of OPCs undergoing apoptosis in the hindbrain. We also observed effects on myelination, marked by an apparent decrease in thickness of the myelin sheath, abnormal patterning of myelinated axon tracts in the optic tectum, and abnormal intensity of white matter regions on MRI. However, because mutants can have a decrease in axon diameter, we cannot conclude definitively that myelin sheath thickness is decreased (for example, see *Klingseisen et al., 2019*). To fully investigate whether the zebrafish VWM mutants have a decreased G-ratio (decreased myelin sheath thickness), future studies should analyze this by binning results by axon diameter.

Since a hallmark of VWM is that disease progression can be precipitated by stressors, we tested the ISR and the response to stress in the *eif2b* mutants. We noted that while stress of wild-type zebrafish caused motor deficits, stress in the mutant *eif2b* zebrafish (including heat shock, hypoxia, or thapsigargin) did not worsen further worsen their already impaired motor behavior (data not shown). This suggests that the *eif2b* mutant zebrafish are already 'maximally' stressed at baseline, consistent with the observed activation of the ISR at baseline, and the finding of increased intron 12 retention. ISR induces expression of transcription factors including ATF4 (*Harding et al., 2000*; *Watatani et al., 2008*). ATF4 and other components of the ISR trigger a stress-induced transcriptional program that promotes cell survival during mild or acute conditions, but which causes pathological changes under severe or chronic insults (*Pakos-Zebrucka et al., 2016*).

A poorly understood aspect of VWM is why a minor stressor can precipitate significant white matter loss and neurological deterioration. Our work suggests that this is caused by a feed-forward effect of chronic ISR that cannot be terminated. This occurs by alternative splicing and retention of

intron 12 in EIF2B5 leading to a premature stop codon and expression of a truncated EIF2B5. This was first observed in cancer cell lines in response to hypoxia (*Brady et al., 2017*). The truncated EIF2B5 permits formation of the eIF2B complex and a partial catalytic domain, but is missing the eIF2α interaction domain. This would result in impaired translation mechanism and a decrease in protein synthesis (*Adomavicius et al., 2019*). We demonstrated the pathological role of truncated EIF2B5 in VWM disease by showing that truncated *eif2b5* is expressed in VWM mutants, and that expression of the truncated EIF2B5 impairs motor function, and itself leads to further activation of the ISR.

Major advances in the past several years have shown that a small molecule stabilizing the decameric eIF2B complex (ISRIB - ISR InhiBitor) can boost the GEF activity of wild-type and VWM mutant eIF2B complexes (*Tsai et al., 2018*; *Wong et al., 2018*) and can inhibit the ISR in mice bearing an $Eif2b5^{R132H/R132H}$ VWM mutation (*Wong et al., 2019*). However, it is not clear whether ISRIB or other drugs targeting eIF2B complex stability will be sufficient to prevent stress-induced ISR chronic, feed-forward activation. Therapies for VWM may need a combinatorial approach incorporating eIF2B stabilization at baseline; and prevention of stress-induced truncated eIF2B5 expression.

# Materials and methods

## Key resources table

| Reagent type (species) or resource | Designation | Source or reference | Identifiers | Additional information |
|---|---|---|---|---|
| Gene (*Danio rerio*) | *eif2b1* | *D. rerio* genome resource | GRCz11: ENSDARG00000091402 | |
| Gene (*D. rerio*) | *eif2b2* | *D. rerio* genome resource | GRCz11: ENSDARG00000041397 | |
| Gene (*D. rerio*) | *eif2b3* | *D. rerio* genome resource | GRCz11: ENSDARG00000018106 | |
| Gene (*D. rerio*) | *eif2b4* | *D. rerio* genome resource | GRCz11: ENSDARG00000014004 | |
| Gene (*D. rerio*) | *eif2b5* | *D. rerio* genome resource | GRCz11: ENSDARG00000074995 | |
| Gene (*D. rerio*) | *apoeb* | *D. rerio* genome resource | GRCz11: ENSDARG00000040295 | |
| Gene (*Homo sapiens*) | EIF2B2 | *H. sapiens* genome resource | GRCh38.p13: ENSG00000119718 | |
| Gene (*H. sapiens*) | EIF2B5 | *H. sapiens* genome resource | GRCh38.p13: ENSG00000145191 | |
| Genetic reagent (*D. rerio*) | $eif2b1^{sa12357}$ | Zebrafish International Resource Center (ZIRC) | ZIRC Catalog ID: ZL8971.02; RRID:ZIRC_ZL8971.02 | |
| Genetic reagent (*D. rerio*) | $eif2b2^{sa17223}$ | Zebrafish International Resource Center (ZIRC) | ZIRC Catalog ID: ZL10609.15; RRID:ZIRC_ZL10609.15 | |
| Genetic reagent (*D. rerio*) | $eif2b4^{sa17367}$ | Zebrafish International Resource Center (ZIRC) | ZIRC Catalog ID: ZL10621.10; RRID:ZIRC_ZL10621.10 | |
| Genetic reagent (*D. rerio*) | $eif2b5^{zc102}$ | This paper- generated and described in Materials and methods. | | |
| Genetic reagent (*D. rerio*) | $eif2b5^{zc103}$ | This paper- generated and described in Materials and methods. | | |
| Genetic reagent (*D. rerio*) | Tg (*ß-actin*:EIF2B2-*2A-eGFP*) | This paper- generated and described in Materials and methods. | | |
| Genetic reagent (*D. rerio*) | Tg(*olig2:dsRed*)$^{vu19}$ | It is available from EZRC and China ZRC | ZFIN ID:ZDB-ALT-080321–2 | |
| Antibody | anti-acetylated tubulin (mouse monoclonal) | Sigma | Sigma: T6793; RRID:AB_477585 | (1:250) |

*Continued on next page*

*Continued*

| Reagent type (species) or resource | Designation | Source or reference | Identifiers | Additional information |
|---|---|---|---|---|
| Antibody | Anti-GFP (mouse monoclonal) | Millipore | Millipore: MAB3580; RRID:AB_94936 | (1:250) |
| Antibody | Anti-GFP (mouse polyclonal) | Aves Labs | Aves Labs : GFP-1020; RRID:AB_10000240 | (1:1000) |
| Antibody | Anti-HuC/D (mouse monoclonal) | Thermo Fisher Scientific | Thermo Fisher Scientific: A-21271; RRID:AB_221448 | (1:400) |
| Antibody | Anti-dsRed | Takara | Clontech: 632496; RRID:AB_10013483 | (1:250) |
| Antibody | Cy-3 anti-rabbit (goat polyclonal) | Millipore | Millipore: AP132C; RRID:AB_92489 | (1:400) |
| Antibody | Alexa 488 donkey anti-mouse (donkey polyclonal) | Thermo Fisher Scientific | Molecular Probes: A-21202; RRID:AB_141607 | (1:400) |
| Antibody | Alexa 633 donkey anti-rabbit (donkey polyclonal) | Thermo Fisher Scientific | Molecular Probes: A-21070; RRID:AB_2535731 | (1:400) |
| Antibody | Alexa 488 goat anti-chicken (goat polyclonal) | Thermo Fisher Scientific | Molecular Probes: A-11039; RRID:AB_2534096 | (1:400) |
| Antibody | Alexa 555 rabbit anti-goat (rabbit polyclonal) | Thermo Fisher Scientific | Molecular Probes: A-21431; RRID:AB_2535852 | (1:400) |
| Peptide, recombinant protein | Gateway BP Clonase II enzyme mix | Thermo Fisher Scientific | Thermo Fisher Scientific: 11789020 | |
| Peptide, recombinant protein | Gateway LR Clonase II enzyme mix | Thermo Fisher Scientific | Thermo Fisher Scientific: 11701020 | |
| Strain, strain background (*Escherichia coli*) | 10-beta | New England BioLabs | NEB: C3019H | |
| Strain, strain background (Human cancer cell line) | SH-SY5Y | Gift from the Pulst lab | RRID:CVCL_0019 | |
| Recombinant DNA reagent | pCR4.0-TOPO | Thermo Fisher Scientific | Thermo Fisher Scientific: 450030, cat # | |
| Recombinant DNA reagent | pDONR P4-P1R | Tol2 kit | Tol2kit.genetics.utah.edu | |
| Recombinant DNA reagent | pDONR 221 | Tol2 kit | Tol2kit.genetics.utah.edu | |
| Recombinant DNA reagent | pDONR P2R-P3 | Tol2 kit | Tol2kit.genetics.utah.edu | |
| Recombinant DNA reagent | Tg (*ß-actin*:EIF2B2-*2A-eGFP*) | This paper- generated and described in Materials and methods. | | |
| Recombinant DNA reagent | Tg (*ß-actin*:EIF2B5-*2A-eGFP*) | This paper- generated and described in Materials and methods. | | |
| Recombinant DNA reagent | pCS2$^+$-Transposase | This paper- generated and described in Materials and methods. | | |
| Transfected construct (Human) | Tg (*ß-actin*:EIF2B2-*2A-eGFP*) | This paper- generated and described in Materials and methods. | | |
| Biological sample (Human) | Human patient sample (serum) | This paper- generated and described in Materials and methods. | | |
| Commercial assay or kit | DIG RNA Labeling Kit (SP6/T7) | Roche | Roche: 11175025910 | |
| Commercial assay or kit | ApopTag-fluorescein | Millipore | Millipore: S7111 | |

*Continued on next page*

*Continued*

| Reagent type (species) or resource | Designation | Source or reference | Identifiers | Additional information |
|---|---|---|---|---|
| Commercial assay or kit | ApopTag-Red | Millipore | Millipore: S7165 | |
| Commercial assay or kit | ImmPACT (TM) VECTOR (R) RED AP Substrate | Vector Laboratories | Vector Laboratories: SK-5105 | |
| Commercial assay or kit | Black Gold II kit | Millipore | Millipore: AG105 | |
| Chemical compound, drug | Applied Biosystems PowerUp SYBR Green Master Mix | Thermo Fisher Scientific | Thermo Fisher Scientific: A25776 | |
| Software, algorithm | MATLAB | Mathworks | Version R2017b | |
| Software, algorithm | EthoVision-XT software | Noldus (https://www.noldus.com/ethovision-xt) | RRID:SCR_000441 | |
| Software, algorithm | GraphPad Prism software | GraphPad Prism (https://graphpad.com) | RRID:SCR_015807 | Version 8.4.3 |
| Software, algorithm | ImageJ software | ImageJ (http://imagej.nih.gov/ij/) | RRID:SCR_003070 | |
| Other | DAPI stain | Sigma | D9542 | (1 µg/mL) |
| Other | ZEG and ZEG chips | wFluidx, Inc | http://www.wfluidx.com | |

## Fish stocks and embryo raising

Adult fish were bred according to standard methods. Embryos were raised at 28.5℃ in E3 embryo medium and staged by time and morphology. For in situ staining and immunohistochemistry, embryos were fixed in 4% paraformaldehyde (PFA) in 1xPBS overnight (O/N) at 4℃, washed briefly in 1xPBS with 0.1% Tween-20, serially dehydrated, and stored in 100% MeOH at −20℃ until use. Transgenic fish line and allele used in this paper was the following: Tg(*olig2:dsRed*)[yu19] (*Kucenas et al., 2008*). *eif2B* Sequence Analysis.

Human and zebrafish *eif2b1*, *eif2b2*, *eif2b3*, *eif2b4* and *eif2b5* subunit genes amino acid sequences were compared using Clustal Omega (*Sievers et al., 2011*) and aligned using PRALINE (*Simossis and Heringa, 2005*); phylogenetic tree analysis was performed with Phylodendron.

## Zebrafish *eif2b* mutant lines obtained from Sanger zebrafish mutation project

We obtained the *eif2b1*[sa12357], *eif2b2*[sa17223], and *eif2b4*[sa17367] alleles generated by the Sanger Zebrafish Mutation Project per our request and established these lines as viable stock at our facility. The *eif2b1*[sa12357] allele results in a T > A nonsense mutation resulting in an early stop in exon 8. The *eif2b2*[sa17223] and *eif2b4*[sa17367] alleles have mutations in an essential splice site predicted to interrupt splicing from exon 5 to 6 of *eif2b2* by a G > A mutation at the end of exon 5 (744 nt), and from exon 12 to 13 in *eif2b4* by a G > A mutation at the end of exon 12 (1404 nt).

## *eif2b5* CRISPR sgRNA construction and injection

Targeting the *D. rerio eif2b5* gene (Ensembl Zv10: ENSDART00000110416.4) for CRISPR mutagenesis was performed by the University of Utah Mutation Generation and Detection Core. We designed sgRNA target sites by looking for sequences corresponding to $N_{18}GG$ on the sense or antisense strand of the DNA using the CRISPR design program at http://crispr.mit.edu. Off-target effects were checked through the use of NIH BLAST tool applied to the zebrafish genome (zv10). Off-target sequences that had significant matches of the final 23 nt of the target and NGG PAM sequence were discarded.

sgRNAs targeting exon 1 (*eif2b5*-ex1) were transcribed using DraI-digested gRNA expression vectors as templates, and the HiScribe T7 RNA Synthesis kit (New England BioLabs) followed by RNA purification with Micro Bio-spin six columns (BioRad).

To maximize mutagenesis and minimize lethality, an *eif2b5-ex1* sgRNA dose curve was performed. A mix of *eif2b5-ex1* sgRNA (between 250 pg-600 pg) and Cas9 protein (600 pg, Integrated DNA Technologies) were injected into one-cell stage embryos, as previously described. CRISPR efficiency was evaluated on individual 24 hpf injected embryos after DNA extraction, PCR amplification of the target locus, and HRMA analysis. An *eif2b5-ex1* sgRNA dose of 450 pg resulted in >90% mutagenesis in 24 hpf embryos, assayed by HRMA, with no difference in survivability compared to uninjected controls. Embryos used for injection were derived from wild-type AB parents, previously confirmed at the *eif2b5* locus to have exact homology for the sgRNA sequence.

## HRMA PCR

DNA was extracted from embryos (24 hpf-72 hpf), or fin clips from adults, into 50 mM NaOH, and incubated at 95 ˚C for 20 min, followed by neutralization with 1M Tris pH 8.0 (1:10 by volume) as previously described (*Xing et al., 2014*). Oligonucleotides were designed using Primer3 to give 60–100 nucleotide products spanning the CRISPR cleavage site, and tested in silico using uMELT to determine PCR products with optimal thermodynamic profiles. PCR was performed using LightScanner Master Mix System (BioFire) in 10 μl reactions as previously described (*Xing et al., 2014*). Thermal melt profiles were collected on a LightScanner (Idaho Technology) (65–98˚C, hold 62˚C) and analyzed with LightScanner Software.

Genotyping was performed by PCR and high-resolution melt analysis (HRMA), using the LightScanner Master Mix system. For *eif2b5* alleles with mutations in exon 1, we used the following primers and conditions: (*eif2b5* forward primer) 5'- AAGCCGGTGTCGGATAAAGAT-3' and (*eif2b5* reverse primer) 5'- AAACCTGCGGTTGAAACTGTC-3'; 95 ˚C for 2 min, followed by 29 cycles of 94 ˚C for 30 s, and 60 ˚C for 30 s, followed by a final denaturation and annealing step of 95 ˚C for 30 s, and 25 ˚C for 30 s. For *eif2b2*[sa17223] genotyping we did PCR with the following primers and conditions: (*eif2b2* forward primer) 5'- TAATGTCACGAGTCAATAAG-3' and (*eif2b2* reverse primer) 5'- AGGATTAATCTTTTATTTCA-3'; 95 ˚C for 2 min, followed by 29 cycles of 94 ˚C for 30 s, and 60 ˚C for 30 s, followed by a final denaturation and annealing step of 95 ˚C for 30 s, and 25 ˚C for 30 s. For *eif2b4*[sa17367] genotyping we did PCR with the following primers and conditions (*eif2b4* forward primer) 5'-TTGAGCATCAAAAGGGTTATTG −3' and (*eif2b4* reverse primer) 5'-CGGACTCTTTTGTATCCAATG −3'; 95 ˚C for 2 min, followed by 29 cycles of 94 ˚C for 30 s, and 55 ˚C for 30 s, followed by a final denaturation and annealing step of 95 ˚C for 30 s, and 25 ˚C for 30 s. All PCR was run in an optically transparent plate with 25 μl mineral oil overlay. We then performed HRMA to differentiate the *eif2b* melt-curves from their corresponding controls.

For *eif2b1*[sa12357] genotyping, HRMA analysis was not able to reliably identify mutant versus wild-type alleles. We therefore used the restriction endonuclease sequence Hpy188I (taTCAGAtg) present in the wild-type allele of *eif2b1*, but not in the *eif2b1*[sa12357] mutant allele (taACAGAtg) to create a restriction fragment length polymorphism (RFLP) for genotyping. We first performed PCR of the target locus (667 bp) in a 10 μl reaction, with the following primers and conditions (*eif2b1* forward primer) 5'- GGAGACGTAAAATGTACCTGCAAT-3' and (*eif2b1* reverse primer) 5'-CACCCCAACCATCACAGGAG-3'; 98 ˚C for 1 m, followed by 34 cycles of 98 ˚C for 10 s, 58 ˚C for 15 s, and 72 for 25 s, followed by a final extension step of 72 ˚C for 10 m. Following PCR, the entire 10 μl reaction was digested with Hpy188I in a 15 μl reaction for 2 hr at 37 ˚C. The digest was then run out on a 2.5% agarose gel. Animals homozygous for *eif2b1*[sa12357/sa12357] produced a 623 bp mutant band, and a 44 bp band unrelated to the *eif2b1*[sa12357] allele, used as an internal control for the restriction digest. Wild-type animals, harboring the non-mutated Hpy188I restriction site, generate a 404 bp and 219 bp band, specific to the wild-type allele, as well as the 44 bp internal control band; heterozygous animals generate all four bands after Hpy188I digestion: mutant 623 bp, wild-type 404 bp and 219 bp, and the internal digest control band 44 bp.

## Chip genotyping

Survival genotyping of 72 hpf zebrafish was performed as previously described (*Lambert et al., 2018*). Briefly, embryos were loaded individually onto genotyping wells of a ZEG chip (wFluidx, Inc) in 12 μl of E3 with a wide bore pipette tip. Once embryos are loaded, the ZEG chip was placed on the vibration unit, covered with the lid and vibrated for 7.5 min at 1.4 volts, 0.026 amps and 0.03 watts. After the samples are vibrated, 10 μl of E3 is removed from the well into a PCR strip tube.

The corresponding embryo is removed from the chip well with a transfer pipette and a small amount of E3, into a 96 square well plate (650 µl/well) until after genotyping. For genotyping PCR, 5 µl of E3 is removed from each well is used directly in the PCR without further cell lysis steps.

Cloning and characterization of zebrafish *eif2b5* CRISPR mutants *eif2b5-ex1* CRISPR injected F0 embryos were raised and crossed to non-mutagenized wild-type AB siblings. *eif2b5* F1 founder offspring were then raised to adulthood, fin-clipped for DNA extraction and PCR amplification of the target locus, using the above described HRMA primers for *eif2b5*. Individual fish identified as potential mutants by HRMA were further confirmed by Topo-TA cloning (Thermo Fisher Scientific) of the target locus, and Sanger sequencing four clones per animal. To ensure that we did not miss larger mutations at the CRISPR mutagenesis site, PCR amplification of a larger region surrounding the target locus (517 bp) was performed using the following primers and PCR conditions: (*eif2b5* forward sequencing primer) 5'- AGCTACCTCAACAGGGCGTA-3', and (*eif2b5* reverse sequencing primer) 5'- CGTCCAAAAACAAAACAGCA-3'; 98 °C for 1 m, followed by 34 cycles of 98 °C for 10 s, 60 °C for 15 s, and 72 for 20 s, followed by a final extension step of 72 °C for 10 m.

## Cloning of human *EIF2B2* and *EIF2B5*

Human *EIF2B2* and *EIF2B5* were amplified from cDNA prepared from SH-SY5Y human cell line. The following primers, and PCR conditions were used to amplify *EIF2B2* for cloning into a middle entry clone Gateway-compatible vector (*EIF2B2* forward primer contains: 31 nt *att*B1F sequence, a nine nt zebrafish optimized Kozak sequence, and 20 nt region of *EIF2B2* beginning at the ATG) 5'- GGGGA-CAAGTTTGTACAAAAAAGCAGGCTACGCCGCCACCATGCCGGGATCCGCAGCGAA-3', and (*EIF2B2* reverse primer contains: 30 nt *att*B2R sequence, and a 20 nt region of the 3' *EIF2B2* that does not contain the stop codon) 5'- GGGGACCACTTTGTACAAGAAAGCTGGGTCTAAAACATGATCATCAGGAT-3'; PCR conditions: 98 °C for 1 m, followed by 34 cycles of 98 °C for 10 s, 68 °C for 15 s, and 72 for 30 s, followed by a final extension step of 72 °C for 10 m. The following primers, and PCR conditions were used to amplify *EIF2B5* for Gateway cloning (*EIF2B5* forward primer contains: 31 nt *att*B1F sequence to allow for Gateway pME cloning, a nine nt zebrafish optimized Kozak sequence, and a 20 nt region of *EIF2B5* beginning at the ATG) 5'- GGGGACAAGTTTG TACAAAAAAGCAGGCTACGCCGCCACCATGGCGGCCCCTGTAGTGGC-3', and (*EIF2B5* Reverse primer contains: 30 nt *att*B2R sequence to allow for Gateway pME cloning, and a 20 nt region of the 3' *EIF2B5* that does not contain the stop codon) 5'-GGGGACCACTTTGTACAAGAAAGCTGGGTCTCAGTCATCTTCAGATGACT-3'; PCR conditions: 98 °C for 1 m, followed by 34 cycles of 98 °C for 10 s, 68 °C for 15 s, and 72 for 75 s, followed by a final extension step of 72 °C for 10 m.

The *att*B containing *EIF2B* PCR product were combined with a donor vector (pDonor #221) in a BP recombination reaction to generate Gateway middle clones, pME *EIF2B2* and pME *EIF2B5*. These plasmids were then diagnostically digested and sequenced to confirm the correct cloning. Expression clones were assembled using the Tol2 kit and recombination reactions with Gateway plasmids. For expression and visualization of human *EIF2B5* (pME *EIF2B5*) in the zebrafish we used the ubiquitous *Beta actin* enhancer containing a minimal core promoter (mcp) encoding the viral E1b TATA box fused to the carp b-actin 5' UTR (p5E *Beta actin*) and the viral 2A bicistronic eGFP fluorescent tag (p3E *2A:eGFP*) assembled into the pDestTol2pA2 plasmid.

## Immunohistochemistry

Immunohistochemistry was performed as previously described (*Bonkowsky et al., 2008*). Antibodies used were: mouse anti-acetylated tubulin 1:250 (Sigma), mouse monoclonal anti-GFP 1:250 (Millipore), chicken anti-GFP 1:1000 (Aves Labs), mouse anti-HuC/D 1:400 (ThermoFisher), rabbit anti-dsRed 1:250 (Clontech), Cy-3 anti-rabbit 1:400 (Millipore), Alexa 488 donkey anti-mouse 1:400, Alexa 633 donkey anti-rabbit, Alexa 488 donkey anti-chicken 1:400, Alexa 555 rabbit anti-goat 1:400 (ThermoFisher), and 4',6-diamidino-2-phenylindole (DAPI).

## In situ *hybridization*

The antisense digoxigenin-labeled cRNA probes for zebrafish *eif2b1-5* and *apoeb* were prepared by using a clone-free method, as previously described (*Thisse and Thisse, 2008*). Forward and reverse primers were generated using Primer3 to generate a PCR product between 800 and 950 nt. The following primers were used: *eif2b1* forward in situ primer 5'- CGTTGCATCAGCGACACTAT-3', *eif2b1*

reverse in situ primer 5'-GAAATGCTTTATAAACAGCAATAATCA-3'; *eif2b2* forward in situ primer 5'-CGCAGGTGAACTGATGGAG-3', *eif2b2* reverse in situ primer 5'-GATGTTTTGAATGCCAGACG-3'; *eif2b3* forward in situ primer 5'-GAGAACAGCGGGACTTTGTC-3', *eif2b3* reverse in situ primer 5'-TTCGTCTTCAGGCCTGTTCT-3',; *eif2b4* forward in situ primer 5'-GGATCCAATGCTCGATCTGT-3', *eif2b4* reverse in situ primer 5'- GAGGGAAGTGTGTGCATCTG-3'; *eif2b5* forward in situ primer 5'-TGGTGGTGGGTCCAGATATT-3', *eif2b5* reverse in situ primer 5'- CAGCCCGTTGTATTTTCCAG-3'; *apoeb* forward in situ primer 5'- ATGAGGTCTCTTGTGGTATTCTTTGCTC-3', *apoeb* reverse in situ primer 5'- TTAAGCCTGAGTGGGAAGAGCTGCTG-3'. To generate a PCR product for antisense probe generation, the reverse primer also contains a 32 nt sequence containing; a 9 nt 5' spacer sequence, a 17 nt T7 polymerase sequence, and a 6 nt 3' spacer (5'-CCAAGCTTCTAATACGAC TCACTATAGGGAGA −3'), resulting in a PCR product <1 kb. The constructs of *olig1* and *mag* ribop-robes are kindly provided by Yi-Chuan Cheng. Synthesis of digoxigenin-labeled antisense RNA probes were synthesized using a DIG RNA Labeling Kit (SP6/T7; Roche).

## Microscopy and image analysis

Immunostained embryos were transferred step-wise into 80% glycerol/20% PBST, mounted on a glass slide with a #0 coverslip, and imaged on a confocal microscope. Confocal stacks were projected in ImageJ, and images composed with Adobe Photoshop and Illustrator.

## TUNEL quantification

Terminal deoxynucleotidyl transferase dUTP nick-end labeling (TUNEL) was performed on whole-mount larvae (ApopTag-Fluorescein In Situ Apoptosis Detection Kit; Millipore) as previously described (*Lambert et al., 2012*). Confocal imaging was performed and images were rendered in ImageJ by compiling a max sum projection of 100 µm (step size 2.5 µm) into a single z-stack image, for cell counting using Photoshop's (Adobe) count tool.

## RNA isolation and cDNA synthesis

RNA was isolated from between 50–100 embryos per sample, depending on age. Each sample was suspended in 900 µl Trizol Reagent (Invitrogen), triturated with a 25-gauge needle until homogeneous; 270 µl chloroform was added and the sample was centrifuged for 15 m at 4 ℃. The aqueous solution was moved to a new tube, an equal volume isopropanol (approximately 500 µl) was added, and 5 µl Glycoblue (Invitrogen) added, followed by centrifugation for 15 m at 4 ℃. The pellet was washed in 70% ethanol, centrifuged for 10 m at 4 ℃, and resuspended in 44 µl DEPC $H_2O$. To remove DNA, 5 µl DNase I buffer and 1 µl Turbo DNase (Invitrogen) was added to each sample and incubate for 15 m at 25 ℃. The volume was brought to 400 µl with DEPC $H_2O$, and an equal volume phenol:chloroform was added, mixed, and centrifuged for 15 m at 4 ℃. The aqueous phase was transferred to a new tube and 1.5 µl Glycoblue was added, 1/10 vol 3M NaOAc added, and 2.5x of the total volume of 100% ethanol was added. The sample was allowed to precipitate overnight at −20 ℃. The sample was then centrifuged for 15 m at 4 ℃, supernatant removed, and the pellet washed in 70% ethanol. This was followed by a final centrifugation step of 15 m at 4 ℃, supernatant removed, and the RNA resuspended in 15 µl DEPC $H_2O$.

First-strand cDNA was made from 1 to 5 µg total RNA using the SuperScript III First-Strand Synthesis System (Applied Biosystems) per manufacturer's instructions.

## qRT-PCR

All qRT-PCR reactions were performed using SYBR Green PCR master mix (Invitrogen) and 2 µl cDNA with the following conditions: 50 ℃ for 2 min, 95 for 10 min followed by 39 cycles of 95 ℃ for 20 s, 60 ℃ for 20 s, and 72 for 20 s, followed by a final melt curve step that ramps in temperature between 60℃ and 95℃.

## Behavior analysis

Larval behavior analysis was performed on 7 dpf larvae in 96-well square bottom plates (Krackeler Scientific) using video analysis software (Noldus EthoVision). For spontaneous behavior, animals were transferred at 6 dpf to the 96-well plate and kept at 28.5 ℃ overnight. At 7 dpf the plate was

placed on the video imaging system and animals were allowed to adapt in the dark for 10 min, and then recording was performed for 5 m (1 min dark and 4 min light).

## TEM

Following fixation overnight at 4 ˚C (in 2.5% glutaraldehyde; 1% PFA in 0.1M sodium cacodylate, 8 mM CaCl$_2$, 2.4% sucrose; pH 7.4), then processed and embedded in plastic as follows: rinsed 2 X ten minutes in 0.1M sodium cacodylate buffer containing 2.4% sucrose and 8 mM CaCl$_2$. Tissue was then post-fixed in 2% osmium tetroxide in a 0.1M sodium cacodylate buffer for 1 hr at room temperature, followed by a rinse for 5 min in water filtered through a 0.22 μm millipore filter. Staining was performed *en bloc* for 1 hr at room temperature with saturated aqueous uranyl acetate, that was filtered through a 0.22 μm millipore filter. Samples were dehydrated through a graded series of ethanol, 10 min at each step at room temperature.

Tissue was then transitioned through three changes of absolute acetone, 10 min each, followed by infiltration with with increasing concentrations of plastic (Embed 812) as follows: 1 hr, Plastic:Acetone 1:1; overnight, Plastic:Acetone 3:1, done at room temperature. Final plastic infiltration was carried out the following day by changing the plastic three times, then placing tissue vials on a rotator for 1 hr and then under vacuum for 1 hr. After the third change tissue was embedded in a fresh plastic in mold with appropriate labels and placed in 60–70˚C oven overnight. Once the plastic was cured the samples were thick sectioned (0.5–1.0 um) and placed on glass slides; stained with 1% Toluidine Blue-O in 1% borax on a hot plate. Tissue was sectioned with a diamond knife on a Leica EMUC6 ultramicrotome, picked up on 150 mesh copper grids and contrasted sequentially with saturated aqueous uranyl acetate followed by staining with Reynold's lead citrate. Sections were examined on a JEOL 1400+ electron microscope.

## MRI

The brains of wild-type (n = 6; 7-months old) and *eif2b5$^{zc103}$* fish (n = 6; 5.5-months old) were dissected leaving the skull, a small piece of the spinal cord, and a small amount of fat tissue surrounding the brain intact. They were placed in 4% paraformaldehyde and 0.5% magnevist (gadopentetate dimeglumine) overnight at 4 ˚C on a nutator.

MR imaging was performed with a 1.0 cm diameter loop-gap radiofrequency transmitter-receiver and a preclinical 7T MRI scanner (Bruker Biospec) with the microimaging gradient set (1100mTm$^{-1}$, BGA-6, Bruker). Two primary scans were done to map the T1 and T2 behavior of the tissue. These scans were 3D turbo spin-echo scans (Rapid Acquisition with Refocused Echoes, RARE), specialized with varying repetition time (TR) or echo time (TE) to optimize T1 and T2 contrast. The T1 map scan utilized the following parameters: T1 weighted RARE: TR = [50, 150, 275, 450, 600]ms, TE = 12 ms, four averages, RARE factor = 2, matrix size = 305×140 x 165, spatial resolution = 0.036 mm isotropic, field of view: 110 x 50 x 60 mm. T2 weighted RARE: TR = 500 ms, TE = [12.2, 36.6, 60.94]ms, nine averages, RARE factor = 2, matrix size = 305 × 140 x 165, spatial resolution = 0.036 mm isotropic, field of view: 110 x 50 x 60 mm. Six zebrafish skulls were scanned in wild-type/mutant pairs in the custom coil.

The raw intensity data was exported from Paravision 5.1 (Bruker BioSpec). This data was imported into MATLAB r2017b (Mathworks, Natick MA), and reshaped into two sets of 3D volumes. For the T1 weighted scan, each volume was representative of an individual TR. For the T2 weighted scans, each volume was representative of a different TE. With this, a gradient descent algorithm could be applied to the intensity vs TR, TE curve of each pixel, for the T1, T2 volumes respectively. General parameters were T1: Initial constants: M0 = 8000, T1 = 150, Loop limit = 75000, learning rate = 0.000001. Equation: $S = M_0(1 - e^{TR/T1})$ T2: Initial constants: M0 = 8000, T2 = 75, Loop limit = 75000, learning rate = 0.000001. Equation: $S = M_0 e^{TE/T2}$.

Sub-millimeter movement was witnessed between the T1 experiments, which caused pixel intensities to not properly fit to the T1 equation, and blurred in the averaged image. This was corrected for with simple 2D rigid body transformations between TR experiments, using the MATLAB r2017b 'affine2d' function. A transformation matrix was generated for a single representative slice, then applied to all slices within the volume. This transformation was applied to each T1 experiment of a given set and were averaged into a single volume.

Transformation matrices were generated with landmark based analysis. Three primary landmarks were utilized to build the transformation: the center of the lens of each eye and the tip of the vagal lobe. The repetitions were averaged and fitted with the gradient descent algorithm detailed above. This resulted in two volumes of T1 and T2 values for each scan pair. The skull pair was separated into two distinct volumes for each T1, T2 map, and rotated into the anatomical planes for visualization.

Analysis of the T1 and T2 maps involved a two-step process of quantifying overall brain volume, and determining the T1 and T2 values for white matter regions of interest (ROI). To begin, four representative measurements were taken for each wild-type and mutant pair. Measurements taken were the length from the tip of olfactory bulb to end of the tectum, the maximum width of the optic tectum, and the maximum height of brain at center ventricle. These measurements were normalized to the width of the specimen's skull.

To measure the intensity of white matter regions in the T2 images, a $2 \times 2$ pixel region was taken from the periventricular grey zone and normalized to a $2 \times 2$ pixel region in the optic tectum from a slice at the end of the rhombencephalic ventricle (RV) of midbrain. Two more slices were measured moving rostrally from the RV.

## Black/gold staining

Adult brains were dissected, fixed overnight in 4% PFA, then incubated 1 hr in 5% sucrose in PBS, overnight in 15% sucrose in PBS, and overnight in 30% sucrose in PBS, prior to embedding in OCT. Tissues were sectioned at 30 µm at −20 ˚C in the cryostat and mounted on positively charged slides.

Black gold II powder (Millipore) was resuspended in 0.9% NaCl to a final concentration of 0.3% and preheated to 60˚C along with 1% sodium thiosulfate. Tissue sections were brought to room temperature for 5 min, then post-fixed in 10% formalin for 1 hr. Slides were incubated at 60˚C in the pre-warmed Black Gold II solution for 15 min. Slides were rinsed 1X in tap water for 2 min, then incubated in sodium thiosulfate solution (1%) for 3 min at 60˚C. After another tap water rinse, slides were dehydrated to 100% EtOH, incubated in 50:50 Hemo-De:EtOH for 3 min, and finally in Hemo-De for 3 min prior to mounting in Cytoseal 60.

## Statistical analysis and blinding

Statistical analyses were performed using Prism6 software (GraphPad). Student's *t*-test was used for two-way comparisons; comparisons between three or more groups was performed with ANOVA with post-hoc Tukey's HSD between individual means.

Samples were randomly allocated to control or experimental groups, other than required distribution by genotype (e.g. wild-type embryos were in the wild-type group). Allocation of samples and animals, data collection, and analysis were performed blinded to genotype and/or experimental group.

# Acknowledgements

We thank Yi-Chuan Cheng for the *olig1* and *mag* constructs, Electron microscopy was performed at the University of Utah Electron Microscopy Core Laboratory with assistance from N Chandler and D Belnap. Confocal imaging was performed at the University of Utah Fluorescence Microscopy Core Facility, supported in part by an NCRR Shared Equipment Grant # 1S10RR024761-01. The University of Utah Centralized Zebrafish Animal Resource (CZAR) provided zebrafish husbandry, and is supported in part by NIH grant # 1G20OD018369-01. Sequencing was performed at the University of Utah DNA Sequencing Core Facility. CRISPR design and construction was performed at the University of Utah Mutation Generation and Detection Core.

# Additional information

## Competing interests

Joshua L Bonkowsky: Consultant: Bluebird Bio (5/2017; 10/2017; 11/2019) Calico (1/2018-1/2019) Denali therapeutics (6/2019) Enzyvant (6/2019) Neurogene (3/2020) Board of Directors wfluidx 1/

2018-present Stock Orchard Therapeutics. The other authors declare that no competing interests exist.

## Funding

| Funder | Grant reference number | Author |
|---|---|---|
| National Institutes of Health | 1R21 NS109441-01 | Joshua L Bonkowsky |
| University of Utah | Bray Chair | Joshua L Bonkowsky |

The funders had no role in study design, data collection and interpretation, or the decision to submit the work for publication.

## Author contributions

Matthew D Keefe, Conceptualization, Data curation, Formal analysis, Supervision, Validation, Investigation, Visualization, Writing - original draft, Writing - review and editing; Haille E Soderholm, Conceptualization, Data curation, Formal analysis, Validation, Investigation, Visualization, Methodology, Writing - original draft, Writing - review and editing; Hung-Yu Shih, Kathryn A Glaittli, D Miranda Bowles, Data curation, Formal analysis, Investigation, Writing - review and editing; Tamara J Stevenson, Conceptualization, Data curation, Formal analysis, Supervision, Investigation, Methodology, Project administration, Writing - review and editing; Erika Scholl, Data curation, Formal analysis, Supervision, Writing - original draft, Project administration, Writing - review and editing; Samuel Colby, Data curation, Formal analysis, Investigation, Methodology; Samer Merchant, Data curation, Formal analysis, Supervision, Investigation, Methodology; Edward W Hsu, Conceptualization, Resources, Formal analysis, Supervision, Methodology, Project administration; Joshua L Bonkowsky, Conceptualization, Resources, Formal analysis, Supervision, Funding acquisition, Methodology, Writing - original draft

## Author ORCIDs

Joshua L Bonkowsky (iD) https://orcid.org/0000-0001-8775-147X

## Ethics

Human subjects: Human subjects-related aspects of the study were approved by the Institutional Review Board of the University of Utah and the Privacy Board of Intermountain Healthcare. Informed consent was obtained including consent to publish, protocol #19596.

Animal experimentation: Zebrafish experiments were performed in strict accordance of guidelines from the University of Utah Institutional Animal Care and Use Committee (IACUC), regulated under federal law (the Animal Welfare Act and Public Health Services Regulation Act) by the U.S. Department of Agriculture (USDA) and the Office of Laboratory Animal Welfare at the NIH, and accredited by the Association for Assessment and Accreditation of Laboratory Care International (AAALAC).

## Decision letter and Author response

Decision letter https://doi.org/10.7554/eLife.56319.sa1
Author response https://doi.org/10.7554/eLife.56319.sa2

# Additional files

## Supplementary files

• Transparent reporting form

## Data availability

All data generated or analysed during this study are included in the manuscript and supporting files. Source data files have been provided for Figures 1, 2, 3, 4, 5, and 6.

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
