## [Decision Letter]

**Acceptance summary:**

In this study, Keefe et al., develop and validate a novel zebrafish model to study the mechanisms underlying Vanishing White Matter disease (VWM). They demonstrate that distinct mutations in multiple eIF2B complex subunits recapitulate several features of VWM, including disrupted growth, impaired myelination and motor defects. They show conservation of function by rescuing some of these defects with a human EIF2B2 ortholog. By leveraging this model, the authors show that the integrated stress response (ISR) that is activated in these mutants results in truncation of the *eif2b5* subunit transcript. This truncated form is capable of causing motor defects and activating the ISR in wild type mice, suggesting that a feed-forward mechanism drives ISR activation in VWM. These findings address could provide insight regarding how minor stressors can lead to significant clinical deterioration and white matter damage in VWM patients.

**Decision letter after peer review:**

Thank you for submitting your article "Vanishing white matter disease expression of truncated EIF2B5 activates induced stress response" for consideration by *eLife*. Your article has been reviewed by three peer reviewers, including Beth Stevens as the Reviewing Editor and Reviewer #1, and the evaluation has been overseen by Richard White as the Senior Editor. The following individual involved in review of your submission has agreed to reveal their identity: Cody J Smith (Reviewer #3).

The reviewers have discussed the reviews with one another and the Reviewing Editor has drafted this decision to help you prepare a revised submission.

As the editors have judged that your manuscript is of interest, but as described below that additional experiments are required before it is published, we would like to draw your attention to changes in our revision policy that we have made in response to COVID-19 (https://elifesciences.org/articles/57162). First, because many researchers have temporarily lost access to the labs, we will give authors as much time as they need to submit revised manuscripts. We are also offering, if you choose, to post the manuscript to bioRxiv (if it is not already there) along with this decision letter and a formal designation that the manuscript is 'in revision at *eLife*'. Please let us know if you would like to pursue this option. (If your work is more suitable for medRxiv, you will need to post the preprint yourself, as the mechanisms for us to do so are still in development.)

Summary

Mutations in the five subunits of the eukaryotic initiation factor 2B (eIF2B1-5) complex cause a severe leukodystrophy called Vanishing White Matter (VWM) disease. In this study from the Bonkowsky lab, new mutants were either generated or imported and analyzed for CNS defects relevant to VWM. By studying growth, lethality, myelination, CNS cell development, and swimming behavior, the authors conclude that the new *eif2b* mutants phenocopy VWM patients. The authors also show that in mutants, a retained intron leads to expression of a truncated transcript, which they conclude acts in a dominant-negative fashion and suggest that this explains some pathology in human VWM patients.

Although modeling human disease in model organisms like zebrafish is important, there are several major issues with the study that dampen enthusiasm as outlined below.

Essential revisions:

Significance of Model

1) For most of five the mutations most of them are heterozygous. Here the authors showed that only 2 out of 4 subunit mutants (*eif2b5*, *eif2b2*) exhibited phenotypes. Is it possible that other mutant alleles have mild phenotypes, such as increased ISR? More characterization of phenotypes in the other mutants (*eif2b1*, *eif2b4*) and the heterozygous siblings is needed to address relevance of this as a model of VWM.

Motor defects

1) It is unconvincing that the movement measurements in *eif2b5^zc103^* mutants represent motor behavior deficits. Please address possible non-specific developmental delays cause the observed phenotypes. Can the authors perform the behavioral experiments at later stages or provide stage matched (rather than age matched) characterizations the mutants?

2) In Figure 6, for the truncated EIF2B5 mis-expression experiments, it is unclear from the text and Materials and methods how the experiments were performed. If the truncated protein acts as a dominant-negative in the e1f2b mutants, why would expression of the wild-type gene rescue the mutant phenotype? The dominant-negative product would still be present. Also, Figure 6D: Compare motor behavior in actin:*eif2b* vs *eif2b5* expression and clarify and address alternative interpretations of the data (See reviewer 3 comments).

Characterization of oligodendrocytes and myelin:

1) Olig2 also labels motor neurons in the spinal cord and neural precursors in the hindbrain. Based on this marker alone it cannot be concluded they are only quantifying oligodendrocyte lineage cells in Figure 3. Please address.

2) Figure 3P-S: the TUNEL staining overlaps with Olig2 in the hindbrain but not in the rest of the brain. Please investigate which other cell types are undergoing apoptosis the optic nerve and optic tectum where the myelination and axon defects are evident in the *eif2b5^zc103^*/103 mutants/

3) Figure 3F-I:the *eif2b2* and *eif2b5* mutants show a striking change in proliferation pattern at 5dpf, including a loss of proliferation in the eyes and cerebellum and increased proliferation in the ventricles. To investigate which cell types are undergoing altered proliferation co-staining for cell-type-specific markers (eg: microglia, astrocytes, neurons) Is needed. See reviewer 1 comments.

4) The electron micrographs in Figure 4 are low quality and cannot be analyzed for G-ratio based on what is shown. Please address.

Reviewer #1:

In the manuscript entitled "Vanishing white matter disease expression of truncated EIF2B5 activates induced stress response", Keefe et al., develop and validate a novel zebrafish model to study the mechanisms underlying Vanishing White Matter disease (VWM). They demonstrate that distinct mutations in multiple *eIF2B* complex subunits recapitulate several features of VWM, including disrupted growth, impaired myelination and motor defects. Remarkably, they show conservation of function by rescuing these defects with a human EIF2B2 ortholog. By leveraging this model, the authors show that the integrated stress response (ISR) that is activated in these mutants results in truncation of the *eif2b5* subunit transcript. This truncated form is capable of causing motor defects and activating the ISR in wild type mice, suggesting that a feed-forward mechanism drives ISR activation in VWM. These findings address a key question in the field regarding how minor stressors can lead to significant clinical deterioration and white matter damage in VWM patients.

Overall, this study presents a potential new model to interrogate VWM pathophysiology and identify therapeutic agents. Below are a few comments that would strengthen the findings.

1) In Figure 3F-I, the *eif2b2* and *eif2b5* mutants show a striking change in proliferation pattern at 5dpf, including a loss of proliferation in the eyes and cerebellum and increased proliferation in the ventricles. The authors are encouraged to investigate which cell types are undergoing altered proliferation by co-staining for cell-type-specific markers (eg: microglia, astrocytes, neurons). This would further inform the origin of the defects seen in the adult *eif2b5^zc103^*/103 mutants.

2) Similarly, in Figure 3P-S, the TUNEL staining overlaps with Olig2 in the hindbrain but not in the rest of the brain. The authors are encouraged to investigate which other cell types are undergoing apoptosis, particularly in the optic nerve and optic tectum where the myelination and axon defects are evident in the eif2b5 zc103/103 mutants.

Reviewer #2:

Mutations in the five subunits of the eukaryotic initiation factor 2B (eIF2B1-5) complex cause a severe leukodystrophy called Vanishing White Matter (VWM) disease. In this study from the Bonkowsky lab, new mutants were either generated or imported and analyzed for CNS defects relevant to VWM. By studying growth, lethality, myelination, CNS cell development, and swimming behavior, the authors conclude that the new *eif2b* mutants phenocopy VWM patients. The authors also show that in mutants, a retained intron leads to expression of a truncated transcript, which they conclude acts in a dominant-negative fashion and suggest that this explains some pathology in human VWM patients. Although modeling human disease in model organisms like zebrafish is important, there are several issues with the study that dampen enthusiasm.

1) In human patients, mutations in each of the 5 eIF2B subunits can cause VWM, and most of them are heterozygous (van der Knaap et al., 2002). In this study, the authors showed that only 2 out of 4 subunit mutants (*eif2b5*, *eif2b2*) exhibited phenotypes. It is possible that other mutant alleles have mild phenotypes, such as increased ISR? Can the authors characterize such phenotypes in the other mutants (*eif2b1*, *eif2b4*) and also in the heterozygous siblings?

2) It is unconvincing that the movement measurements in *eif2b5^zc103^* mutants represent motor behavior deficits. The authors point out that these mutants are smaller than wild-type controls and the mutant larvae fail to inflate their swim bladders. Thus, it is also possible that non-specific developmental delays cause the observed phenotypes. Can the authors perform the behavioral experiments at later stages or provide stage matched (rather than age matched) characterizations the mutants?

3) Olig2 also labels motor neurons in the spinal cord and neural precursors in the hindbrain. Thus, the authors cannot conclude they are only quantifying oligodendrocyte lineage cells in Figure 3.

4) The electron micrographs in Figure 4 are low quality and cannot be analyzed for G-ratio based on what is shown. These images look more like toluidine blue stained semi-thick sections than EM sections.

5) In Figure 6, for the truncated EIF2B5 mis-expression experiments, it is unclear from the text and Materials and methods how the experiments were performed. Did the authors generate a stable transgenic line, as for EIF2B2, or it was a transient injection of expressing DNA construct? If it was a transient injection experiment, expression would be mosaic and likely short-term; thus, a motor phenotype is surprising if this was the method.

6) If the truncated protein acts as a dominant-negative in the e1f2b mutants, why would expression of the wild-type gene rescue the mutant phenotype? The dominant-negative product would still be present.

Reviewer #3:

Keefe and Soderholm and Shih et al. present a thorough paper that uses multiple approaches to address how *eif2b5* functions in neural development and function. The work nicely evaluated common features of VWM disease and sets foundation for future studies that would dissect mechanistic regulators in VWM. The model and data that is present will be of broad interest to groups studying neural development as well as VWM. The conclusions that are stated in the paper nicely match the data that is presented. Overall, this is a well written and complete story.

In Figure 6D, the impaired motor movement is observed. The authors conclude that it is acting as a dominant negative. However, it could be over expression and not a dominant negative effect. Can the authors compare motor behavior in actin:*eif2b* vs *eif2b5* expression? Alternatively, can the authors add text that demonstrate alternative interpretations of the data.

[Editors' note: further revisions were suggested prior to acceptance, as described below.]

Thank you for resubmitting your work entitled "Vanishing white matter disease expression of truncated EIF2B5 activates induced stress response" for further consideration by *eLife*. Your revised article has been evaluated by Richard White (Senior Editor) and a Reviewing Editor.

The manuscript has been improved but there are some remaining issues that need to be addressed before acceptance, as outlined below. Overall, this manuscript can serve as a useful model for understanding how mutations in eIF2b2 can leading to phenotypes present in VWM and will be a resource for the community. The authors have included several new data panels that help to clarify some of the previous concerns. However, a few issues remain that need to be addressed before publication.

1) A major issue remains in the quality of the electron microscopy to show myelin ultrastructure. As shown, it is unclear how g-ratio could have been calculated from images shown. Please address comments below (reviewer 1) and provide data using high quality EMs images as is standard for the field and important for a manuscript focused on a model of Vanishing White Matter disease.

2) New data in supplemental figures 1 and 2 are critical are important for the main story and should be included in the main figures for the story as written and revised. However, it is difficult to visualize the co-labeling of tunel and ph3 with the noted cell-type markers in those images that influence the quantification of data and results.

3) Please provide clarification and details on the sox10:mrfp transgene used in the study: The authors listed sox10:mrfp as a transgene that they used in the study but it is not clear which figure that transgenic animal was used. Tg(sox10:mrfp) would be a useful tool to validate the OPC phenotype in the spinal cord since sox10 labels OPCs with very different morphology than the subset of neurons that are labeled with that transgene.

Reviewer #2:

The authors were decently responsive to the previous round of critiques. A major issue remains in the quality of the electron microscopy to show myelin ultrastructure. As shown, it is unclear how g-ratio could have been calculated from images shown. Additionally, g-ratio quantifications are usually binned by axon size, which was not performed here and likely cannot be based on the quality of the EMs. The EMs are not publication quality and should not be published. If publication quality EMs cannot be obtained and quantified, the histological analysis of myelin will have to suffice, although this is certainly not ideal for a manuscript focused on a model of Vanishing White Matter disease. In this case, myelin was never formally examined, and the wording throughout the Abstract and manuscript will need to be changed to reflect this.

Reviewer #3:

Overall, this manuscript can serve as a useful model for understanding how mutations in eIF2b2 can leading to phenotypes present in VWM and will be a resource for the community. The authors have provided a point by point rebuttal to the reviewers concern. The authors have also included several new data panels that help to clarify some of the previous concerns. Most of this new data matches the conclusions as written. However, some of new data shown has presented a concern or clarification that should be addressed before publication.

New data in supplemental figures 1 and 2 are critical for the story as written and revised. It is difficult to visualize the co-labeling of tunel and ph3 with the noted cell-type markers in those images. It is not clear how such quantification could be completed, especially with the resolution of in situ hybridization vs immunostaining. As written, these figures are important for the main story and should be included in the main figures. The tunel staining in these new supplementary files is also different than displayed in the main figures. Can the authors provide insets for the labeling demonstrating an example of a co-labeled cell in each of the genotypes.

In an effort for the authors to address a review regarding the specificity of cell death in the spinal cord they used mag as a marker for oligodendrocytes. mag is certainly one marker that could identify an oligodendrocyte. But the mag in situ appears to be present in areas of the PNS without defined myelinating cells. There are also a low number of mag+ cells in the spinal cord of animal at 5 dpf in those images than would be expected for labeling of myelinating cells. Can the authors test that the mag+ cells in the spinal cord are also olig2+ or sox10+ to confirm that the mag labeling is specific.

---

## [Author Response]

Reviewer #1:In the manuscript entitled "Vanishing white matter disease expression of truncated EIF2B5 activates induced stress response", Keefe et al., develop and validate a novel zebrafish model to study the mechanisms underlying Vanishing White Matter disease (VWM). They demonstrate that distinct mutations in multiple eIF2B complex subunits recapitulate several features of VWM, including disrupted growth, impaired myelination and motor defects. Remarkably, they show conservation of function by rescuing these defects with a human EIF2B2 ortholog. By leveraging this model, the authors show that the integrated stress response (ISR) that is activated in these mutants results in truncation of the eif2b5 subunit transcript. This truncated form is capable of causing motor defects and activating the ISR in wild type mice, suggesting that a feed-forward mechanism drives ISR activation in VWM. These findings address a key question in the field regarding how minor stressors can lead to significant clinical deterioration and white matter damage in VWM patients.Overall, this study presents a potential new model to interrogate VWM pathophysiology and identify therapeutic agents. Below are a few comments that would strengthen the findings.1) In Figure 3F-I, the eif2b2 and eif2b5 mutants show a striking change in proliferation pattern at 5dpf, including a loss of proliferation in the eyes and cerebellum and increased proliferation in the ventricles. The authors are encouraged to investigate which cell types are undergoing altered proliferation by co-staining for cell-type-specific markers (eg: microglia, astrocytes, neurons). This would further inform the origin of the defects seen in the adult eif2b5^zc103^/103 mutants.

We have now examined this and have found that, interestingly, that proliferation was dysregulated across cell types, and affected glia, neurons, and microglia; and also had spatial specificity.

2) Similarly, in Figure 3P-S, the TUNEL staining overlaps with Olig2 in the hindbrain but not in the rest of the brain. The authors are encouraged to investigate which other cell types are undergoing apoptosis, particularly in the optic nerve and optic tectum where the myelination and axon defects are evident in the eif2b5 zc103/103 mutants.

We have now provided new data to address this. We found that the apoptosis only affected glial lineage cells, and only in the midline.

Reviewer #2:Mutations in the five subunits of the eukaryotic initiation factor 2B (eIF2B1-5) complex cause a severe leukodystrophy called Vanishing White Matter (VWM) disease. In this study from the Bonkowsky lab, new mutants were either generated or imported and analyzed for CNS defects relevant to VWM. By studying growth, lethality, myelination, CNS cell development, and swimming behavior, the authors conclude that the new eif2b mutants phenocopy VWM patients. The authors also show that in mutants, a retained intron leads to expression of a truncated transcript, which they conclude acts in a dominant-negative fashion and suggest that this explains some pathology in human VWM patients. Although modeling human disease in model organisms like zebrafish is important, there are several issues with the study that dampen enthusiasm.1) In human patients, mutations in each of the 5 eIF2B subunits can cause VWM, and most of them are heterozygous (van der Knaap et al., 2002). In this study, the authors showed that only 2 out of 4 subunit mutants (eif2b5, eif2b2) exhibited phenotypes. It is possible that other mutant alleles have mild phenotypes, such as increased ISR? Can the authors characterize such phenotypes in the other mutants (eif2b1, eif2b4) and also in the heterozygous siblings?

We thank the reviewer for this question. To clarify one point- in VWM, the disease is autosomal recessive.

We have further characterized the mutants with quite extensive experimental additions; including providing heterozygous data; and for the eif2b4 mutant, to demonstrate that our findings are consistent in a different subunit. Unfortunately, our eif2b1 mutant is quite sick, and we have difficulty getting enough offspring to analyze (we are just trying to keep the line alive). We analyzed survival, growth, behavior, and ISR qRT-PCR, and have included this in Figure 2—source data 3-6.

Given that we have 9 different lines, we selectively chose three alleles for the majority of our experiments. Since all four alleles we characterized in-depth showed similar phenotypes, and which is similar to the human disease, we are reassured that the phenotypes we are observed are representative of human disease.

Finally we have been trying very hard to get zebrafish mutants that have the exact same mutation as the human disease, but, in zebrafish generating allele-specific knock-ins is still quite difficult, and we have been pursuing several strategies for several years without success at this point. For future work in VWM disease this will be important, a point we have added to the Discussion. The human allele mutant in zebrafish will be really helpful to understand the degree of ISR, for example.

2) It is unconvincing that the movement measurements in eif2b5^zc103^ mutants represent motor behavior deficits. The authors point out that these mutants are smaller than wild-type controls and the mutant larvae fail to inflate their swim bladders. Thus, it is also possible that non-specific developmental delays cause the observed phenotypes. Can the authors perform the behavioral experiments at later stages or provide stage matched (rather than age matched) characterizations the mutants?

This is a good point. The phenotypes co-segregate with the mutations, but we do appreciate that the behavioral effects with our data are unable to differentiate a CNS effect, from an effect caused by the size of the animal, etc. The zc103 mutants stay persistently smaller throughout their life span, so stage matching is not a possibility. To clarify a point- the zc103 do have swim bladder (zc102 does not). We discuss this limitation on interpretation of the findings in the text.

3) Olig2 also labels motor neurons in the spinal cord and neural precursors in the hindbrain. Thus, the authors cannot conclude they are only quantifying oligodendrocyte lineage cells in Figure 3.

That is a good point about the Olig2:dsRed transgenic line. We clarify this point, but do also point out that the *olig2:dsRed* line is specific to OPCs in labeled cells that have migrated dorsally out of the spinal cord.

However, to more conclusively test this issue, we have now down experiments using in situ probes for oligodendrocytes (olig1 and mag), and we confirm that there is an increase in apoptosis of oligodendrocytes.

4) The electron micrographs in Figure 4 are low quality and cannot be analyzed for G-ratio based on what is shown. These images look more like toluidine blue stained semi-thick sections than EM sections.

We have included high resolution images of the TEM sections for Figure 4.

5) In Figure 6, for the truncated EIF2B5 mis-expression experiments, it is unclear from the text and Materials and methods how the experiments were performed. Did the authors generate a stable transgenic line, as for EIF2B2, or it was a transient injection of expressing DNA construct? If it was a transient injection experiment, expression would be mosaic and likely short-term; thus, a motor phenotype is surprising if this was the method.

We apologize that this was unclear- the experiment was performed with transient injection. With transient injections, because of the high copy number of plasmids, it is not infrequent to obtain a strong response, which is sometimes more marked than the response from a stable transgenic line, depending on the enhancer. We have clarified this in the text.

6) If the truncated protein acts as a dominant-negative in the e1f2b mutants, why would expression of the wild-type gene rescue the mutant phenotype? The dominant-negative product would still be present.

The reviewer raises an important point, it is possible that the phenotype observed is related to overexpression, and not from a dominant-negative effect. We have provided clarification in the text.

Reviewer #3:Keefe and Soderholm and Shih et al. present a thorough paper that uses multiple approaches to address how eif2b5 functions in neural development and function. The work nicely evaluated common features of VWM disease and sets foundation for future studies that would dissect mechanistic regulators in VWM. The model and data that is present will be of broad interest to groups studying neural development as well as VWM. The conclusions that are stated in the paper nicely match the data that is presented. Overall, this is a well written and complete story.In Figure 6D, the impaired motor movement is observed. The authors conclude that it is acting as a dominant negative. However, it could be over expression and not a dominant negative effect. Can the authors compare motor behavior in actin:eif2b vs eif2b5 expression? Alternatively, can the authors add text that demonstrate alternative interpretations of the data.

This is a good point, and we have revised our text to match the alternative potential explanation.

[Editors' note: further revisions were suggested prior to acceptance, as described below.]

1) A major issue remains in the quality of the electron microscopy to show myelin ultrastructure. As shown, it is unclear how g-ratio could have been calculated from images shown. Please address comments below (reviewer 1) and provide data using high quality EMs images as is standard for the field and important for a manuscript focused on a model of Vanishing White Matter disease.

We have provided images that are at higher magnification, that were used for the quantification and the g-ratio calculations. We have included these images in the revised Figure 6, and updated the text and figure legend. The images are similar in magnification and quality to those routinely published in the field (e.g. Ackerman et al., 2014, Nat Comm; Strachan et al., 2017, Hum Mol Genet; etc.), including for Vanishing White Matter disease (e.g. Dooves et al., 2016; Klok et al., Ann Clin Transl Neurol, 2018; etc.).

2) New data in supplemental figures 1 and 2 are critical are important for the main story and should be included in the main figures for the story as written andrevised. However, it is difficult to visualize the co-labeling of tunel and ph3 with the noted cell-type markers in those images that influence the quantification ofdata and results.It is not clear how such quantification could be completed, especially with the resolution of in situ hybridization vs immunostaining. As written, these figures are important for the main story and should be included in the main figures. The tunel staining in these new supplementary files is also different than displayed in the main figures. Can the authors provide insets for the labeling demonstrating an example of a co-labeled cell in each of the genotypes.

For Figure 3 (the quantification of TUNEL and apoptosis in the different alleles): We have now included the images in the main body of the manuscript, as new Figures 4 and 5, and re-numbered the subsequent figures. The differences in appearances of some of the images are because it is now more than a year after the original images were generated, so new experiments were performed. For the new Figure 4, showing co-labeling of TUNEL and olig1: we have also provided high-magnification insets showing examples of double-labeled cells for each genotype.

In an effort for the authors to address a review regarding the specificity of cell death in the spinal cord they used mag as a marker for oligodendrocytes. mag is certainly one marker that could identify an oligodendrocyte. But the mag in situ appears to be present in areas of the PNS without defined myelinating cells. There are also a low number of mag+ cells in the spinal cord of animal at 5 dpf in those images than would be expected for labeling of myelinating cells. Can the authors test that the mag+ cells in the spinal cord are also olig2+ or sox10+ to confirm that the mag labeling is specific.

We have now included a high-magnification image, showing co-localization of the mag in situ probe with GFP expressed from a transgenic *sox10:GFP* line, taken using a single slice confocal image (step size 2.5mm). This image is Figure 4—figure supplement 1.

The mag staining the reviewer notes in the PNS (in the region ventral to the spinal cord) is biological; mag is a marker for oligodendrocytes as well as for Schwann cells. The different appearances of the PNS mag labeling reflects differences in the positioning of the image used for the paper. Also, with regards to TUNEL/OPC quantification, we note that the separate experiments performed in Figure 3, K-O, that the method of quantification of the olig2:dsRed line, of cells that have dorsally migrated in the spinal cord, is specific for OPCs.

3) Please provide clarification and details on the sox10:mrfp transgene used in the study: The authors listed sox10:mrfp as a transgene that they used in the study but it is not clear which figure that transgenic animal was used. Tg(sox10:mrfp) would be a useful tool to validate the OPC phenotype in the spinal cord since sox10 labels OPCs with very different morphology than the subset of neurons that are labeled with that transgene.

We apologize about the confusion. Although we listed this line in the list of reagents, we did not in fact list or use it in the paper or in the figures. This was because although w had used this line in some experiments, those experiments ended up not being included in the submission, and so had erroneously left the line still in the reagents listed as being used, which we have now removed in the text and in the key resource table.